# A Theoretical and Empirical Model of the Generalization Error under Time-Varying Learning Rate

## Abstract

Stochastic gradient descent is commonly employed as the most principled optimization algorithm for deep learning, and the dependence of the generalization error of neural networks on the given hyperparameters is crucial. However, the case in which the batch size and learning rate vary with time has not yet been analyzed, nor the dependence of them on the generalization error as a functional form for both the constant and time-varying cases has been expressed. In this study, we analyze the generalization bound for the time-varying case by applying PAC-Bayes and experimentally show that the theoretical functional form for the batch size and learning rate approximates the generalization error well for both cases. We also experimentally show that hyperparameter optimization based on the proposed model outperforms the existing libraries.

## 1 Introduction

While deep learning addresses a variety of practical tasks, the reason deep learning generalizes well is not fully understood. It is hoped that the performance of deep learning methods can be improved by understanding deep learning. Consequently, deep learning will become a cornerstone for the next generation of technology. Stochastic gradient descent (SGD) is a principled learning algorithm for deep learning, and because most deep learning algorithms are developed based on SGD, its theoretical analysis is crucial. He et al. (2019) modeled the generalization bounds when both the batch size and learning rate are constant in SGD; however, they did not model the generalization error. Furthermore, it is crucial to decrease the learning rate as the training progresses to make SGD converge (Robbins & Monro, 1951), which results in good performance (You et al., 2020), but He et al. (2019) did not consider the case in which these parameters are time-varying. We analyze the time-varying case in SGD and propose a generalization error model for both constant and time-varying cases. Based on these models, we also propose a novel hyperparameter search method. In this study, we utilized a PAC-Bayes-based generalization bound to generate a model of the generalization error. To validate the proposed model, we trained approximately 300 networks on both practical networks (VGG16 (Simonyan & Zisserman, 2015), Wide-ResNet 28-10 (Zagoruyko & Komodakis, 2016)) and data (CIFAR10, CIFAR100 (Krizhevsky et al., 2009)). We also showed that the proposed model can approximate the experimental results well. We also show that the proposed model can be used for hyperparameter search via Bayesian optimization and can yield better results, compared with the previous method for searching in the uniform distribution, and comparable results for searching in the logarithmic uniform distribution.

The contributions of the present study are summarized as follows:

- We analyzed the generalization bound using PAC-Bayes with the time-varying batch size and learning rate.
- We directly modeled the generalization error on the batch size and learning rate and proposed its functional form.
- We experimentally showed that it is possible to model the generalization error with high accuracy for practical data.
- Our experimental results demonstrated that the hyperparameter search method built on the proposed model can outperform current libraries when

searching over uniform distributions and has comparable performance when searching over logarithmic uniform distributions.

## 2 RELATED WORK

In this section, we present related work on SGD modeling, generalization bounds, prediction of model performance, and hyperparameter optimization.

**Theoretical analysis of SGD:** SGD is often treated as a continuous-time stochastic differential equation (SDE) by modeling the noise of the gradient and the loss function and applying the Euler–Maruyama approximation (Lin et al., 2018; Mandt et al., 2017; Li et al., 2017; Jastrzębski et al., 2017). This approximation can be solved analytically as an Ornstein–Uhlenbeck process (Uhlenbeck & Ornstein, 1930). In this study, we also analyzed SGD using the same process as the previously mentioned studies. However, in most previous studies, the batch size and learning rate in SGD were treated as constants. Hence, in this study, we treat them as time-varying and seek an analytic solution.

**Theoretical error bounds:** There is a large body of research on theoretical explanations for the generalization capability of deep neural networks (Negrea et al., 2019; Arora et al., 2018; Cao & Gu, 2019; Deng et al., 2021). He et al. (2019), building on their analysis of SGD, provided a theoretical explanation of the effects of the batch size and learning rate on the generalization error. However, this research only handles the case of constant batch size and learning rate and does not model the generalization error in the function form. In this study, we derive a generalization bound based on an analytic solution for the time-varying case and derive a model of the generalization error.

**Predicting model performance:** Most studies are based on the results from training networks, assuming hyperparameters as inputs, and searching for the best parameters without training the networks. In neural architecture search, the performance of a model was predicted for a given model on a given dataset (Istrate et al., 2019), small data (Klein et al., 2017), and model size (Real et al., 2019), respectively. Several studies predict the performance of a model in a functional form by treating both the dataset and model sizes as input (Rosenfeld et al., 2020), or the pruning scale as input (Rosenfeld et al., 2021). In this study, we exploit the findings of the above studies to predict the performance of a model in a functional form, based on a theoretically derived generalization bound to the batch size and learning rate.

**Hyperparameter optimization:** Hyperparameter optimization algorithms are mainly categorized into two types: nonadaptive and adaptive. Nonadaptive algorithms search for the best hyperparameters without using any past search results and, thus, have the advantage of being easily parallelizable. Grid search and random search fall into this category, and many studies have shown that random search is more useful (Bergstra & Bengio, 2012; Li & Talwalkar, 2020). Adaptive algorithms search for the best hyperparameters by exploiting past search results and can obtain better hyperparameters with fewer searches. Adaptive algorithms include specific algorithms, such as the evolutionary algorithm (Young et al., 2015) and Bayesian optimization (Wu et al., 2019), and practical tools, such as Hyperopt (Bergstra et al., 2015) and Optuna (Akiba et al., 2019). These methods are flexible enough to be optimized for arbitrary functions. However, there is still room for more efficient search when searching only for specific hyperparameters. In this study, we aim to further improve the search efficiency of both the batch size and learning rate by employing a hyperparameter search based on the proposed generalization error model.

In the following sections, we will analyze the influence of the batch size and learning rate on the generalization error in SGD, based on assumptions about the distribution of the gradients and the shape of the risk function near minima, an approximation to SDE, and the PAC-Bayes method.

## 3 PRELIMINARIES

In this section, we describe the preliminary knowledge used to model the learning process of SGD.

**Generalization error of stochastic algorithms.** In general, machine learning algorithms are designed to select a hypothesis function $F_\theta$ from a hypothesis class $\{F_\theta | \theta \in \Theta \subset \mathbb{R}^d\}$ such that the expected value of the risk function $\mathcal{R}$ computed from the loss function $l$ is minimized. In a proba-

bilistic algorithm such as SGD, the parameter $\theta$ after learning is considered to lie on the distribution $Q$. The risk function for parameter $\theta$ is defined as follows:

$$\mathcal{R}(\theta) = \mathbb{E}_{(X,Y)\sim\mathcal{D}}\left[l(F_\theta(X), Y)\right]. \tag{1}$$

Therefore, $\mathcal{R}(Q) = \mathbb{E}_{\theta\sim Q}\left[\mathcal{R}(\theta)\right]$. However, one cannot compute the risk function, and in practice, the empirical risk $\hat{\mathcal{R}}(\theta)$ from the data is approximated as the risk function $\mathcal{R}(\theta)$.

$$\hat{\mathcal{R}}(\theta) = \frac{1}{N}\sum_{i=1}^{N} l(F_\theta(X_i), Y_i), \tag{2}$$

where $(X_i, Y_i)$ is the $N$ training data consisting of the input data and label pairs. Thus, $\hat{\mathcal{R}}(Q) = \mathbb{E}_{\theta\sim Q}[\hat{\mathcal{R}}(\theta)]$.

**Stochastic gradient descent.** Gradient descent is an algorithm that computes the gradient of the risk function defined in Eq.1 and updates the parameter $\theta$ based on the gradient. The gradient $g(\theta(t))$ is defined as $g(\theta(t)) \triangleq \nabla_{\theta(t)}\mathcal{R}(\theta(t)) = \nabla_{\theta(t)}\mathbb{E}_{(X,Y)}\left[l(F_{\theta(t)}(X), Y)\right]$. We update the parameter $\theta$ by $\theta(t+1) = \theta(t) - \eta g(\theta(t))$, where $\eta$ is the learning rate. In practice, the true gradient of $g(\theta(t))$ is approximated by the gradient of $\hat{g}_S(\theta(t))$ computed from the empirical risk $\hat{\mathcal{R}}(\theta)$ defined in Eq.2, and the parameter $\theta$ is updated with $\hat{g}_S(\theta(t))$. Thus, the gradient $\hat{g}_S(\theta(t))$ is defined as $\hat{g}_S(\theta(t)) = \nabla_{\theta(t)}\hat{\mathcal{R}}(\theta(t)) = \frac{1}{|S|}\sum_{i\in S}\nabla_{\theta(t)}l(F_{\theta(t)}(X_i), Y_i)$, and the parameter $\theta$ is updated by $\theta(t+1) = \theta(t) - \eta\hat{g}_S(\theta(t))$. We use a mini-batch $S$ to train the model in parallel, and $|S|$ represents its size. Then, we treat $l_i(\theta) = l(F_{\theta(t)}(X_i), Y_i)$.

**Analysis of the learning process.** The loss function $l_i(\theta)$ and its mean $\hat{\mathcal{R}}(\theta)$ is an unbiased estimator of the risk function $\mathcal{R}(\theta)$, and the gradient of the loss function $\nabla_\theta l_i(\theta)$ and its mean $\hat{g}_S(\theta)$ is an unbiased estimator of the gradient $g(\theta) = \nabla_\theta\mathcal{R}(\theta)$. Hence, we have $\mathbb{E}[l_i(\theta)] = \mathbb{E}[\hat{\mathcal{R}}(\theta)] = \mathcal{R}(\theta)$ and $\mathbb{E}[\nabla_\theta l_i(\theta)] = \mathbb{E}[\hat{g}_S(\theta)] = g(\theta) = \nabla_\theta\mathcal{R}(\theta)$, where the expectations are computed from the data distribution $(X, Y)$. Here, as in Mandt et al. (2017), we assume that $\{\nabla l_i(\theta)\}$ is independently and identically sampled from a Gaussian distribution with mean $g(\theta) = \nabla_\theta\mathcal{R}(\theta)$. We can express $\nabla_\theta l_i(\theta) \sim \mathcal{N}(g(\theta), C)$, where $C$ is the covariance matrix, which is assumed to be a constant matrix for all $\theta$. Then the gradient of $\hat{g}_S(\theta)$ computed for a mini-batch follows the following Gaussian distribution:

$$\hat{g}_S(\theta) = \frac{1}{|S|}\sum_{i\in S}\nabla_\theta l_i(\theta) \sim \mathcal{N}\left(g(\theta), \frac{1}{|S|}C\right). \tag{3}$$

Because the covariance matrix is symmetric and (semi) positive definite, we suppose that $C$ can be factorized as $C = BB^\top$. In an actual SGD algorithm, the parameters are iteratively updated using Eq.3 to minimize the risk function $\mathcal{R}(\theta)$:

$$\Delta\theta(t) = \theta(t+1) - \theta(t) = -\eta\hat{g}_S(\theta(t)) = -\eta g(\theta) + \frac{\eta}{\sqrt{|S|}}B\Delta W, \quad \Delta W \sim \mathcal{N}(0, I). \tag{4}$$

**Analytical solution of the stochastic differential equation.** When the learning rate $\eta$ is small, Eq.4 can be approximated by SDE, which is known as Ornstein–Uhlenbeck process (Uhlenbeck & Ornstein, 1930). Here, we assume that the risk function is convex and second-order differentiable in the neighborhood of the minimum (He et al., 2019). In addition, when the risk function $\mathcal{R}(\theta)$ is minimized at $\theta = \theta^*$, it is shifted so that it is minimized at $\theta = 0$. Thus, the risk function $\mathcal{R}(\theta)$ can be approximated by second-order statistic on the distribution $Q$ as follows:

$$\mathcal{R}(\theta) = \frac{1}{2}\theta^\top A\theta, \tag{5}$$

where $A$ is the Hessian matrix around the minimum of the risk function and is assumed to be positive definite. This assumption was experimentally demonstrated by Li et al. (2018), and other studies have made similar assumptions (Jastrzębski et al., 2017; Poggio et al., 2017). Because $\mathbb{E}[\hat{\mathcal{R}}(\theta)] = \mathcal{R}(\theta)$, we further assume that the empirical risk $\hat{\mathcal{R}}(\theta)$ is in the following quadratic form:

$$\hat{\mathcal{R}}(\theta) = \frac{1}{2}(\theta - \theta_b)^\top \hat{A}(\theta - \theta_b) \tag{6}$$

where $\theta_b$ is the bias term, $\hat{A}$ is the Hessian matrix, and $\mathbb{E}[\theta_b] = 0$, $\mathbb{E}[\hat{A}] = A$.

It is well known that Eq.4 has an analytic solution (Gardiner, 2004),

$$\theta(t) = e^{-At}\theta(0) + \sqrt{\frac{\eta}{|S|}} \int_0^t e^{-A(t-t')}B\mathrm{d}W(t'), \tag{7}$$

where $W(t')$ is white noise and follows $\mathcal{N}(0, I)$ and $Q$ is the distribution of $\theta$ in Eq.7 and Gaussian.

## 4 THEORETICAL GENERALIZATION BOUND

In this section, we derive the generalization bounds for both constant and time-varying cases.

### 4.1 CONSTANT CASE

Using Eq.7 and PAC-Bayes, we obtain a generalization bound for SGD as follows (see Appendix B.1 for the derivation).

**Theorem 1** (extension of He et al. (2019), Theorem 2). *We treat $A$ as the Hessian of the risk function around the local minimum, $C$ as the covariance matrix of the gradients calculated by single sample points, $Q$ as the distribution of the output hypothesis function of SGD, $\Sigma$ is the covariance matrix of the distribution $Q$, $d$ is the dimension of the parameter $\theta$ (network size), and $R$ is the search radius of the parameter $\theta$.*

*When $A$ and $\Sigma$ commute, for any positive real $\delta \in (0, 1)$, with probability $1 - \delta$ over a training sample set of size N, we have the following inequality for the distribution $Q$:*

$$\mathcal{R}(Q) \leq \hat{\mathcal{R}}(Q)$$
$$+ \sqrt{\frac{\frac{\eta}{2|S|}\mathrm{tr}(CA^{-1}) + d\log\left(\frac{2|S|}{\eta}\right) - \log\left(\det\left(CA^{-1}\right)\right) + R^2 - d + 2\log\left(\frac{1}{\delta}\right) + 2\log N + 4}{4N - 2}}. \tag{8}$$

This generalization bound incorporates a new scale factor for the parameter search, $R^2$, into Theorem 2 of He et al. (2019). Although similar results can be derived without assuming that $A$ and $\Sigma$ commute, we used the above assumption for simplicity. $R$ represents the radius of the hypersphere in the parameter search, and $\|\theta - \theta_0\|^2 \leq R^2$. We also show that this generalization bound also has a network initialization factor. A detailed description is provided in Appendix B.2.

### 4.2 TIME-VARYING CASE

It should be noted that, when solving Eq.4, the hyperparameters in SGD such as the batch size $|S|$ and the learning rate $\eta$ can be time-varying. While He et al. (2019) derived a generalization bound for the case in which these are constant, we further analyze the time-varying case. In the time-varying case, the solution takes the form of a hyperparameter term in Eq.7 entering the integral (Gardiner, 2004).

$$\theta(t) = e^{-At}\theta(0) + \int_0^t \sqrt{n(t')}e^{-A(t-t')}B\mathrm{d}W(t'), \tag{9}$$

where $n(t) = \frac{\eta(t)}{|S(t)|}$. Then, we treat the time-varying batch size and learning rate as a univariate function $n(t)$. This allows us to treat changes in the learning rate (Robbins & Monro, 1951; You et al., 2020) and batch size (Smith et al., 2018) in a unified manner. In general, if $n(t)$ is allowed to be an arbitrary function, this integral cannot be solved analytically. However, because the learning rate generally decreases and the batch size generally increases as the learning progresses, if we assume $n(t)$ to be a monotonically decreasing function and apply the mean value theorem for integrals $\int_{t_0}^{t_1} n(t)F(t)\mathrm{d}t = n(\xi) \int_{t_0}^{t_1} F(t)\mathrm{d}t, \xi \in [t_0, t_1]$, then we obtain a generalization bound for the time-varying case as follows (see Appendix B.3 for the detailed derivation).

**Theorem 2.** *We treat $D = \int_0^t e^{-A(t-t')} C e^{-A(t-t')} \mathrm{d}t'$. Using the same settings as theorem 1 and when $A$ and $\Sigma$ commute and $A$ and $D$ commute, we have the following inequality for the time-varying case:*

$$\mathcal{R}(Q) \leq \hat{\mathcal{R}}(Q)$$

$$+ \sqrt{\frac{\frac{n_{inter}}{2}\mathrm{tr}(CA^{-1}) + d\log\left(\frac{2}{n_{inter}}\right) - \log\left(\det\left(CA^{-1}\right)\right) + R^2 - d + 2\log\left(\frac{1}{\delta}\right) + 2\log N + 4}{4N - 2}}, \tag{10}$$

*and*

$$n_{inter} = (1 - \lambda)\min\left(n(t)\right) + \lambda\max\left(n(t)\right), \quad \lambda \in [0, 1]. \tag{11}$$

## 5 FUNCTIONAL APPROXIMATION OF THE GENERALIZATION ERROR

Based on the theoretical analysis thus far, we consider approximating $\mathcal{R}(Q)$ in a functional form. From Eq.8 and Eq.10, we have already modeled the generalization bound $(\mathcal{R}(Q) - \hat{\mathcal{R}}(Q))$. We can also design a empirical functional form based on He et al. (2019)'s experimental results on the effect of ratio of the batch size to learning rate on the generalization error. Thus, if we can model the training error $\hat{\mathcal{R}}(Q)$, we can derive the functional form of the generalization error. We model the training error by substituting Eqs.7 and 9 into Eq.6. Subsequently, we show the specific functional form of the generalized error model for constant and time-varying cases.

### 5.1 CONSTANT CASE

First, we organize Eq.7 with respect to the batch size $|S|$ and learning rate $\eta$ and the learned parameter $\theta(t)$ is as follows:

$$\theta(t) = \boldsymbol{a_0} + \sqrt{\frac{\eta}{|S|}}\boldsymbol{a_1}, \tag{12}$$

where $\boldsymbol{a_0}, \boldsymbol{a_1}$ are vectors that do not depend on $|S|, \eta$. By substituting this into Eq.6, the functional form of the training error $\hat{\mathcal{R}}(Q)$ is as follows. The detailed derivation is in Appendix B.4.1.

$$\hat{\mathcal{R}}(Q) \simeq \mathbb{E}_{\theta \sim Q}\left[\frac{1}{2}(\theta - \theta_b)^\top \hat{A}(\theta - \theta_b)\right] = \mathbb{E}\left[\frac{1}{2}\left(\boldsymbol{a_0'} + \sqrt{\frac{\eta}{|S|}}\boldsymbol{a_1}\right)^\top \hat{A}\left(\boldsymbol{a_0'} + \sqrt{\frac{\eta}{|S|}}\boldsymbol{a_1}\right)\right]$$

$$= a_0 + a_1\frac{\eta}{|S|} + a_2\sqrt{\frac{\eta}{|S|}}, \tag{13}$$

where $\boldsymbol{a_0'} = \boldsymbol{a_0} - \theta_b$, and $a_0, a_1, a_2$ are constants. Although we can model $\mathcal{R}(Q)$ directly by substituting Eq.7 into Eq.5, this cannot be treated as a practical functional form as it does not model the experimental increase of the generalization bound with respect to the decrease of $\frac{\eta}{|S|}$ (He et al., 2019). Thus, we address this problem by modeling the training error $\hat{\mathcal{R}}(Q)$ and the generalization bound $(\mathcal{R}(Q) - \hat{\mathcal{R}}(Q))$ separately. To model the generalization bound, we propose the following functional form based on the model derived in Eq.8 and He et al. (2019)'s experimental results. The detailed derivation is given in Appendix B.5.1.

$$\mathcal{R}(Q) - \hat{\mathcal{R}}(Q) \simeq a_3 + a_4\sqrt{\frac{|S|}{\eta}} \tag{14}$$

Here, $a_3, a_4$ are constants. By combining Eq.13 and Eq.14, we propose the following functional form as a model of the generalization error $\mathcal{R}(Q)$.

$$\mathcal{R}(Q) = \tilde{\epsilon}(S, \eta) = \left(c_0\frac{\eta}{|S|} + c_1\sqrt{\frac{\eta}{|S|}}\right) + \left(c_2\sqrt{\frac{|S|}{\eta}}\right) + c_3 \tag{15}$$

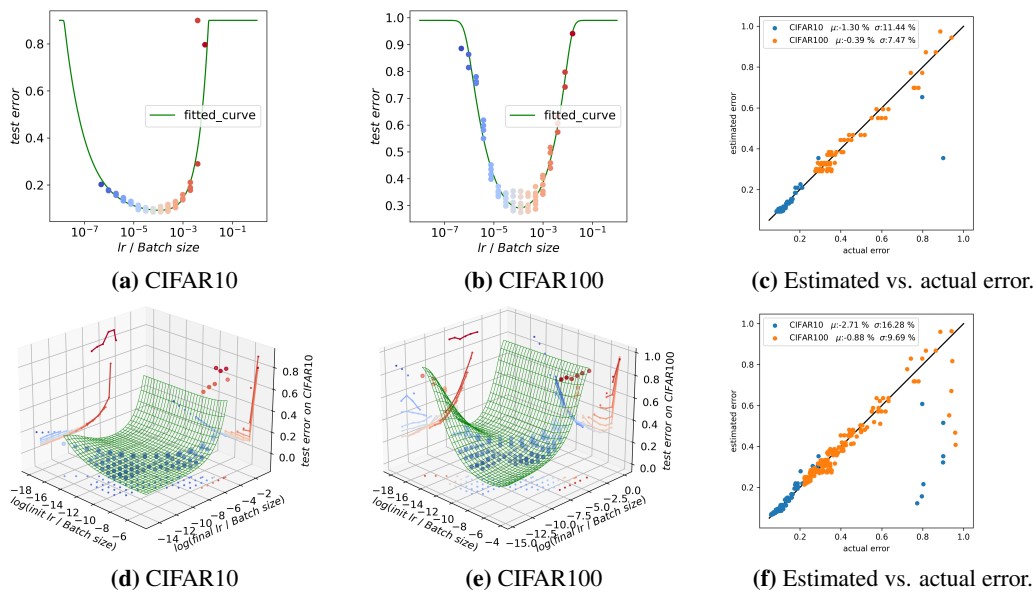

**Figure 1:** Experimental results of the proposed model for both (**a**, **b**, **c**) constant and (**d**, **e**, **f**) time-varying cases. (**a**, **b**) For the constant case, we plotted points with the ratio of the learning rate to the batch size on the horizontal axis of a log scale and the generalization error on the vertical axis. (**d**, **e**) For the time-varying case, we plotted points with the ratio of the initial learning rate to the batch size and the ratio of the final learning rate to the batch size on the horizontal axis of the log scale, and the generalization error on the vertical axis. (**a**, **b**, **d**, **e**) For all cases, we also show the results fitted by the proposed model as green lines and surfaces, respectively. (**c**, **f**) The true generalization error and estimation error are plotted, and the mean and standard deviation of the relative error $\delta$ are represented by $\mu$ and $\sigma$, respectively.

where $c_0 \sim c_3$ are constants. The first term models the training error $\hat{\mathcal{R}}(Q)$, and the second term models the generalization bound $(\mathcal{R}(Q) - \hat{\mathcal{R}}(Q))$. Thus, the first term denotes that the training error monotonically increases with respect to the hyperparameter $\frac{\eta}{|S|}$, and the second term denotes that the generalization bound inversely decreases.

## 5.2 Time-Varying Case

As discussed in Section 4.2, we approximate Eq.9 to a simple form similar to Eq.12, using an approximation of the mean value theorem for integrals. Based on the functional form obtained by substituting Eq.9 into Eq.6, the model of generalization bounds derived in Eq.10, and He et al. (2019)'s experimental results, we propose the following functional form as the model of the generalization error $\epsilon(S, \eta)$. A detailed derivation is given in Appendix B.4.2 and B.5.2.

$$\tilde{\epsilon}(S, \eta) = \left( c_0 n_1^2 + c_1 n_1 \right) + \left( c_2 \sqrt{\frac{1}{n_2}} \right) + c_3 \tag{16}$$

$$n_1 = (1 - \lambda_1) \min \left( \sqrt{n(t)} \right) + \lambda_1 \max \left( \sqrt{n(t)} \right) \tag{17}$$

$$n_2 = (1 - \lambda_2) \min \left( n(t) \right) + \lambda_2 \max \left( n(t) \right) \tag{18}$$

where $c_0 \sim c_3, \lambda_1, \lambda_2$ are constants and $\lambda_1, \lambda_2 \in [0, 1]$. Note that, for $n_1$ in the first term, the intermediate value is computed after taking the square root of $n(t)$, and for $n_2$ in the second term, the linear interpolation is computed.

## 5.3 Practical Functional Form

In the discussion above, we constructed a rough model of the generalization error. However, the models in Eqs.15 and 16 have no upper bounds, and they don't model the fact that the loss function

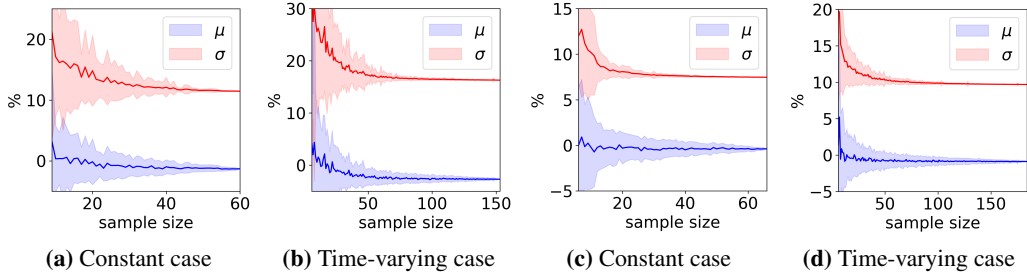

**Figure 2:** Experimental results on the stability of the generalization error model for (**a**, **b**) CIFAR10 and (**c**, **d**) CIFAR100, for the (**a**, **c**) constant case and (**b**, **d**) time-varying case. The horizontal axis shows the number of networks randomly selected to estimate the generalization error, whereas the vertical axis shows the mean value $\mu$ and standard deviation $\sigma$ of the relative error $\delta$. The shaded areas represent one standard deviation from the mean in each direction.

hits its head at the random guess level. Therefore, it is necessary to model the transition from the initial random guess level to the proposed model $\tilde{\epsilon}(S, \eta)$. Thus, we propose the model $\hat{\epsilon}(S, \eta)$ using a soft-min function based on the LogSumExp function.

$$\hat{\epsilon}(S, \eta) = \text{softmin}(\tilde{\epsilon}(S, \eta), \epsilon_0; c_4) = \frac{1}{c_4}\left(-\log\left(e^{-c_4\tilde{\epsilon}(S,\eta)} + e^{-c_4\epsilon_0}\right)\right) \tag{19}$$

where $c_4$ is a constant that controls the shape (sharpness) of the soft-min function ($\text{softmin}(x, y) = -\log\left(\exp\left(-x\right) + \exp\left(-y\right)\right)$). $\epsilon_0$ is the value of the loss function at the random guess level, which is statistically determined and not explored. (e.g., for a balanced dataset in image classification, we can treat it as $\epsilon_0 = 1 - (1/N_{classes})$.)

## 6 EXPERIMENT

To validate the proposed model $\hat{\epsilon}(S, \eta)$, we trained a practical network (VGG16 (Simonyan & Zisserman, 2015), Wide-ResNet 28-10 (Zagoruyko & Komodakis, 2016)) on the image classification tasks (CIFAR10, CIFAR100 (Krizhevsky et al., 2009)).

### 6.1 EXPERIMENTAL SETTINGS AND RESULT

In the constant case, the batch size and learning rate were fixed throughout the training process. To evaluate the generalization error, we trained 60 VGG models with a batch size $|S| \in \{16, 32, 64, 128, 256, 512\}$ and a learning rate $\eta \in \{1/2^i | i = 3, ..., 12\}$ for CIFAR10, as well as 66 Wide-ResNet models with the same batch size $|S|$ and learning rate $\eta \in \{1/2^i | i = 2, ..., 12\}$ for CIFAR100 over a total of 200 epochs. The experimental results for the constant case are shown in Figures.1a and 1b. We plotted the log scale of $\frac{\eta}{|S|}$ on the horizontal axis and the generalization error on the vertical axis. We found both a decreasing and an increasing property of the generalization error before and after $\frac{\eta}{|S|} \simeq 10^{-4}$, which we discussed in Eq.15. Although He et al. (2019) claimed that increasing $\frac{\eta}{|S|}$ decreases the generalization error, the experimental results show that the generalization error increases when $\frac{\eta}{|S|}$ exceeds a certain value. This implies that, in addition to the generalization bound, the training error $\hat{\mathcal{R}}(Q)$ must be modeled.

In the time-varying case, only the learning rate is set to be time-varying, and we use an exponential step function. We chose $\eta_{init}$ and $\eta_{final}$ and trained over 200 epochs, starting with a learning rate of $\eta_{init}$, decaying with a decay rate of $\gamma$ at 60, 120, and 160 epochs, and finally reached a learning rate $\eta_{final} = \gamma^3 \eta_{init}$ at 160 epochs.

In addition to the constant case, we trained a case in which the batch size $|S| \in \{16, 32, 64, 128, 256\}$, the learning rate of $\eta_{init} \in \{1/2^i | i = 3, ..., 4\}$ and $\eta_{final} \in \{0.1/2^i | i = 0, ..., 4\}$ for CIFAR10, the same batch size $|S|$, the learning rate of $\eta_{init} \in \{1/2^i | i = 2, ..., 4\}$ and the same $\eta_{final}$ for CIFAR100, as well as a case in which the batch size $|S| \in \{128, 256\}$, the learning rate of $\eta_{init} \in \{1/2^i | i = 5, ..., 9\}$ and $\eta_{final} \in \{0.1/2^i | i = 5, ..., 9\}$ for both datasets. In

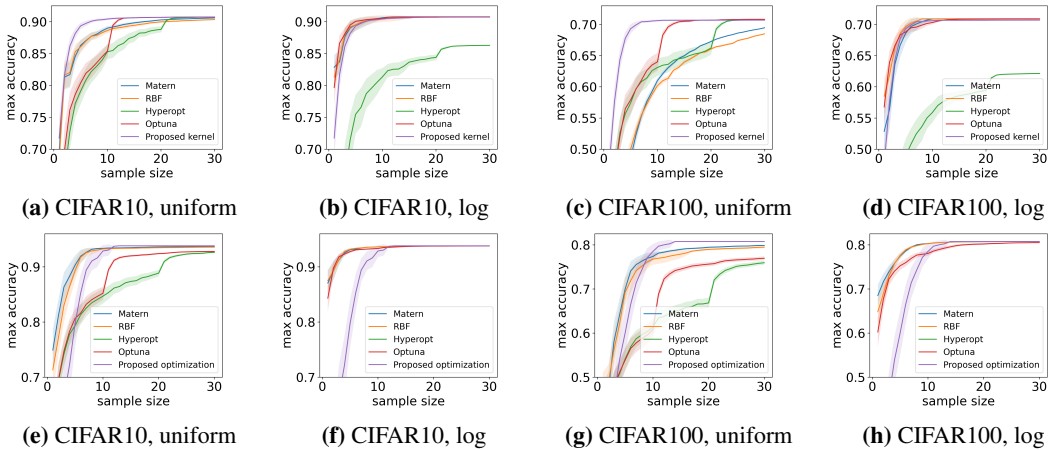

**(a)** CIFAR10, uniform  **(b)** CIFAR10, log  **(c)** CIFAR100, uniform  **(d)** CIFAR100, log

**(e)** CIFAR10, uniform  **(f)** CIFAR10, log  **(g)** CIFAR100, uniform  **(h)** CIFAR100, log

**Figure 3:** Hyperparameter optimization results of several methods. The number of trained models is plotted on the horizontal axis, and the maximum accuracy achieved among the trained models is plotted on the vertical axis for the (**a**, **b**, **c**, **d**) constant and (**e**, **f**, **g**, **h**) time-varying cases. The 95% confidence intervals are shaded. The proposed method always searches for a uniform range, whereas the comparison method searches the range written in the caption (uniform or log-uniform).

total, we measured the generalization error for the 153 models for CIFAR10 and 184 models for CIFAR100. More implementation details can be found in Appendix C.1. The experimental results for the time-varying case are shown in Figures.1d and 1e. $\frac{\eta_{init}}{|S|}$ and $\frac{\eta_{final}}{|S|}$ are plotted on the horizontal axis, whereas the generalization error $\epsilon(S, \eta)$ is plotted on the vertical axis. We also plot the results of projecting the generalization error onto $\frac{\eta_{init}}{|S|}$ and $\frac{\eta_{final}}{|S|}$, respectively. As in the constant case, we can observe that, when the value of $n(t) = \frac{\eta(t)}{|S(t)|}$ is not in the appropriate range (either quite large or quite small), the generalization error becomes large.

## 6.2 ERROR LANDSCAPE ESTIMATION

We estimate the obtained generalization error $\epsilon(S, \eta)$ using a parametric family, $\hat{\epsilon}(S, \eta; \phi)$. Here, we treat the parameter $\phi$ as $\phi = \{c_0, c_1, c_2, c_3, c_4\}$ for the constant case, and then $\{\lambda_1, \lambda_2\}$ is added to the time-varying case. As in Rosenfeld et al. (2021), we search for $\phi$ such that the squared error of $\delta(S, \eta; \phi) = \frac{\hat{\epsilon}(S, \eta; \phi) - \epsilon(S, \eta)}{\epsilon(S, \eta)}$ is minimized. Implementation details are provided in Appendix C.2.

Figure.1c plots our results for the constant case. The mean $\mu$ and standard deviation $\sigma$ of the relative error $\delta$ are $|\mu| < 2\%$ and $\sigma < 12\%$, respectively. Considering the small number of parameters used to estimate the generalization error ($|\phi| = 5$), the proposed functional model can accurately estimate the complex generalization error landscape (60 and 66 models, respectively). In Figure.1f, we plot the results for the time-varying case. We achieved $|\mu| < 3\%$ and $\sigma < 17\%$, respectively. Although the relative error is larger than the constant case, we can still declare that the proposed functional model can estimate the generalization error sufficiently, considering that the generalization error of 153 or 184 models is modeled with a small number of parameters ($|\phi| = 7$). In Appendix C.2.3, we provide a comparison between the proposed model, a model that analytically solves the time-varying case of the step function, and a model that uses only the final learning rate.

In addition, we verified the stability of the proposed model with respect to the number of networks. We randomly sampled a certain number of generalization errors obtained by training the networks, estimated the parameter $\phi$ from them only, and computed the relative error $\delta$. We performed this procedure 100 times and computed $\mu$ and $\sigma$ of the relative error. Details of the experimental setup are presented in Appendix C.3.1.

In Figure.2 we show the experimental results. Figures.2a and 2c show that, for the constant case of CIFAR10 and CIFAR100, we can accurately estimate the generalization error landscape by randomly training about 30 models. In contrast, Figures.2b and 2d show that, for the time-varying case, we need to train about 75 models for the estimation.

## 7 HYPERPARAMETER OPTIMIZATION

In this section, we show that the proposed model is useful for applications such as hyperparameter optimization. In this study, we used Bayesian optimization.

Bayesian optimization can compute a function $y = f(x)$ that can be fitted to any sequence of data points $\{(x_i, y_i)\}$. Nogueira (2014) achieved this by using a Gaussian process with a Matern kernel, and many Bayesian optimization methods use generic kernel functions which can be fitted to any function. While this is practical for fitting unknown functions, ideally, the most accurate Bayesian optimization can be achieved by employing a kernel function constructed from a feature extraction function $\boldsymbol{\psi}(x)$ that can compute the unknown function with a linear combination of $y = \boldsymbol{w}^\top \boldsymbol{\psi}(x)$. Therefore, for the constant case, we propose a kernel function $k(x_i, x_j)$ for the input $x = \frac{\eta}{|S|}$ based on Eq.15 for hyperparameter optimization.

$$k(x_i, x_j) = \boldsymbol{\psi}(x_i)^\top \boldsymbol{\psi}(x_j) = \sqrt{x_i x_j} + x_i x_j + \frac{1}{\sqrt{x_i x_j}} + 1. \tag{20}$$

Details are in Appendix C.4.1. In contrast, the time-varying case cannot be represented by a linear combination of $y = \boldsymbol{w}^\top \boldsymbol{\psi}(x)$, because the product term of the parameters ($c$ and $\lambda$) appears. Therefore, we employ a hyperparameter search by sequential model-based global optimization (SMBO) (Bergstra et al., 2011) with the proposed model in Eq.19 using Eq.16. For SMBO, we use the Levenberg–Marquardt algorithm (Moré, 1978) and Thompson Sampling (Thompson, 1933; Russo et al., 2017). More details are provided in Appendix C.4.2.

To validate the proposed method, we explored $\frac{\eta}{|S|}$ using the fitted generalization error model $\hat{\epsilon}(S, \eta; \boldsymbol{\phi})$ as the training result of the network. The details are given in Appendix C.4.3. Figure.3 shows the experimental results. The horizontal axis plots the number of training networks, and the vertical axis plots the maximum test accuracy among the networks trained to date. A total of 100 experiments were conducted, and the 95% confidence interval was shaded. The methods used for comparison were Bayesian optimization using the Matern kernel and RBF kernel (Nogueira, 2014), Hyperopt (Bergstra et al., 2015), and Optuna (Akiba et al., 2019).

As shown in the Figure.3, in almost all cases, the proposed method achieves the highest accuracy with the least number of models in the search method with uniform distribution. This indicates that the proposed model represents a good feature extractor, and the proposed kernel function is useful for hyperparameter optimization. In addition, in the constant case, the proposed method achieved the highest accuracy with the same number of models as the comparison method, which searches for logarithmic uniform distributions. It indicates that theoretical model-based hyperparameter optimization can be applied practically. In practice, it is generally believed that it is better to search for hyperparameters $S, \eta$ in the logarithmic uniform range (Rasmussen, 1997; Sundararajan & Keerthi, 2001). However, there is no theoretical foundation for this (this is confirmed by the fact that in Hyperopt, the results are not improved by exploring the logarithmic range). These results suggest that the proposed model-based hyperparameter optimization is both theoretically sound and practically useful when performing a hyperparameter search on the batch size and learning rate.

## 8 DISCUSSION AND CONCLUSION

In this paper, we presented a theoretical analysis and a model of the generalization error for the batch size and learning rate in SGD. We analyzed the case in which these parameters are constant during training and the more challenging and practical case when they vary with time. Through experiments for both cases, we show that the proposed generalization error model is practical, and we test the number of networks required to estimate the generalization error landscape (= stability of the generalization error model). In addition, we demonstrate that a hyperparameter search based on the proposed model outperform the conventional Bayesian optimization and hyperparameter optimization libraries, such as Hyperopt and Optuna. To the best of our knowledge, this is the first study that explores the batch size and learning rate based on the generalization error model. From the discussion, we conclude that this theoretical model is a practical and useful model for understanding the effects of the batch size and learning rate on the generalization error. We hope that this work will provide further theoretical insights into deep learning and new research directions for hyperparameter search based on theoretical models.

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

# A    BACKGROUND

## A.1    PAC-BAYESIAN FRAMEWORK

The PAC-Bayesian framework originates from McAllester (1999a;b)'s work. In the PAC-Bayesian perspective, the hypothesis functions learned by probabilistic machine learning algorithms are drawn at random from the hypothesis classes based on some "rules." The generalization ability of an algorithm is negatively correlated with the distance (often measured by the Kullback–Leibler (KL) divergence) between the output hypothesis distribution and the a priori distribution (typically a Gaussian or uniform distribution). The above results suggest that there is a tradeoff between minimizing the empirical risk and exploring further regions of the hypothesis space from the beginning (a priori).

Let $P$ be the initial priori distribution in the parameter space $\Theta$, and let $Q$ be the distribution in $\Theta$ after learning. We can now define the expected value of the risk function under distribution $Q$ as follows:

$$\mathcal{R}(Q) = \mathbb{E}_{\theta \sim Q}[\mathcal{R}(\theta)].$$

Similarly, the empirical risk function under distribution $Q$ can be defined as follows:

$$\hat{\mathcal{R}}(Q) = \mathbb{E}_{\theta \sim Q}[\hat{\mathcal{R}}(\theta)].$$

Based on these, we obtain the classical result in that the expected value of the risk function $\mathcal{R}(Q)$ is uniformly bounded by the expected value of the empirical risk function $\hat{\mathcal{R}}(Q)$ and KL-divergence $\mathcal{D}(Q\|P)$.

**Lemma 1** (See McAllester (1999a), Theorem1). *For any positive real $\delta \in (0,1)$ with a probability of at least $1 - \delta$ over a sample of size $N$, we have the following inequality for all distributions $Q$:*

$$\mathcal{R}(Q) \leq \hat{\mathcal{R}}(Q) + \sqrt{\frac{\mathcal{D}(Q\|P) + \log\left(\frac{1}{\delta}\right) + \log N + 2}{2N - 1}}, \tag{21}$$

*where $\mathcal{D}(Q\|P)$ is the KL divergence between the distributions $Q$ and $P$ and is defined as*

$$\mathcal{D}(Q\|P) = \mathbb{E}_{\theta \sim Q}\left(\log \frac{Q(\theta)}{P(\theta)}\right). \tag{22}$$

# B    SUPPLEMENTAL PROOFS FOR THE PROPOSED METHOD

In this section, we describe the details of the proof, which we omitted in the proposed method.

## B.1    GENERALIZATION BOUND FOR THE CONSTANT CASE

To prove Theorem 1, we use an existing proof method that approximates the SGD update of the model to a stochastic differential equation (see, e.g., (Mandt et al., 2017)). With appropriate assumptions, the learning process of SGD can be approximated by the Ornstein–Uhlenbeck process (Uhlenbeck & Ornstein, 1930). Because the Ornstein–Uhlenbeck process has an analytic solution, we can use the distribution $Q$ of the analytic solution $\theta$ to derive the KL-divergence, which is used to derive a generalization error bound from Eq.21. Here, we modify and extend He et al. (2019)'s Theorems 1 and 2 to derive the generalization error bound for the constant case.

**Lemma 2** (cf. Gardiner (2004), pp.109-110). *Under second-order differentiable assumption (Eq.5), if $t$ is sufficiently large, the Ornstein–Uhlenbeck process (Eq.4)'s distribution*

$$q(\theta) = M \exp\left\{-\frac{1}{2}\theta^\top \Sigma^{-1}\theta\right\}, \tag{23}$$

*has the following property:*

$$A\Sigma + \Sigma A = \frac{\eta}{|S|}C. \tag{24}$$

This lemma is similar to that of Gardiner (2004).

*Proof of Lemma 2.* From the analytical solution of the Ornstein–Uhlenbeck process (Uhlenbeck & Ornstein, 1930), the parameter $\theta$ can be expressed as follows:

$$\theta(t) = e^{-At}\theta(0) + \sqrt{\frac{\eta}{|S|}} \int_0^t e^{-A(t-t')} B \mathrm{d}W(t'), \tag{25}$$

where $W(t')$ is a white noise and follows $\mathcal{N}(0, I)$. From Eq.23, we obtain

$$\Sigma = \mathbb{E}_{\theta \sim Q}\left[\theta\theta^\top\right]. \tag{26}$$

Thus, we obtain the following equation:

$$
\begin{aligned}
A\Sigma + \Sigma A &= Ae^{-At}\langle\theta(0), \theta^\top(0)\rangle e^{-At} + e^{-At}\langle\theta(0), \theta^\top(0)\rangle e^{-At}A \\
&\quad + \frac{\eta}{|S|}\int_0^t Ae^{-A(t-t')}Ce^{-A(t-t')}\mathrm{d}t' \\
&\quad + \frac{\eta}{|S|}\int_0^t e^{-A(t-t')}Ce^{-A(t-t')}A\mathrm{d}t' \\
&= Ae^{-At}\langle\theta(0), \theta^\top(0)\rangle e^{-At} + e^{-At}\langle\theta(0), \theta^\top(0)\rangle e^{-At}A \\
&\quad + \frac{\eta}{|S|}\int_0^t \frac{\mathrm{d}}{\mathrm{d}t'}e^{-A(t-t')}Ce^{-A(t-t')}\mathrm{d}t' \\
&= Ae^{-At}\langle\theta(0), \theta^\top(0)\rangle e^{-At} + e^{-At}\langle\theta(0), \theta^\top(0)\rangle e^{-At}A \\
&\quad + \frac{\eta}{|S|}C - \frac{\eta}{|S|}e^{-At}Ce^{-At} \\
&\simeq \frac{\eta}{|S|}C, \tag{27}
\end{aligned}
$$

where $A$ is the Hessian matrix and symmetric. The last transformation is guided by the assumption that $A$ is a positive definite matrix and all its eigenvalues are positive and by the approximation of the term multiplied by $e^{-At}$ twice with zero, because $t$ is sufficiently large.

The proof is completed. $\qquad\square$

**Lemma 3** (extension of He et al. (2019), Appendix B.1). *For any positive real $\delta \in (0, 1)$, with probability $1 - \delta$ over a training sample set of size N, we have the following inequality for the distribution Q of the output hypothesis function of SGD:*

$$\mathcal{R}(Q) \leq \hat{\mathcal{R}}(Q) + \sqrt{\frac{\frac{\eta}{|S|}\mathrm{tr}(CA^{-1}) - 2\log(\det(\Sigma)) + 2R^2 - 2d + 4\log\left(\frac{1}{\delta}\right) + 4\log N + 8}{8N - 4}}, \tag{28}$$

*where $A$ is the Hessian of the risk function around the local minimum, $C$ is the covariance matrix of the gradients calculated by single sample points, $\Sigma$ is the covariance matrix of the distribution $Q$, $d$ is the dimension of the parameter $\theta$(network size), and $R$ is the search radius of the parameter $\theta$.*

*Proof of Lemma 3.* In the PAC-Bayesian framework ((Lemma 1), it is crucial to compute the KL-divergence between the parameter distribution of the output hypothesis function and the prior parameter distribution in the hypothesis space. The prior distribution can be interpreted as an initial parameter distribution, and in most cases, either a uniform distribution or a Gaussian distribution is used. In this study, we assume that the prior distribution is Gaussian $\mathcal{N}(-\theta^*, I)$. $\theta^*$ is a parameter for which the risk function $\mathcal{R}(\theta)$ is minimized. The mean value $-\theta^*$ of the prior distribution corresponds to the shift in $\theta$ in Eq.5, which actually eliminates prior knowledge of the posterior distribution. He et al. (2019) uses $\mathcal{N}(0, I)$ for the prior distribution. However, it is unnatural because the prior distribution has prior knowledge of the posterior distribution. Conversely, our setting

becomes more natural. The posterior distribution $Q$ and prior distribution $P$ are represented by $p(\theta)$ and $q(\theta)$ with respect to the parameter $\theta$, given by the following equation:

$$p(\theta) = \frac{1}{\sqrt{(2\pi)^d \det(I)}} \exp\left(-\frac{1}{2}(\theta + \theta^*)^\top I(\theta + \theta^*)\right) \tag{29}$$

$$q(\theta) = \frac{1}{\sqrt{(2\pi)^d \det(\Sigma)}} \exp\left(-\frac{1}{2}\theta^\top \Sigma^{-1}\theta\right). \tag{30}$$

Here, $d$ represents the dimension of the parameter $\theta$, and Eq.30 computed the normalization term $M$ in Eq.23. Thus, we obtain

$$\log\left(\frac{q(\theta)}{p(\theta)}\right)$$

$$= \log\left(\sqrt{\frac{(2\pi)^d \det(I)}{(2\pi)^d \det(\Sigma)}} \exp\left(-\frac{1}{2}(\theta + \theta^*)^\top I(\theta + \theta^*) - \frac{1}{2}\theta^\top \Sigma^{-1}\theta\right)\right)$$

$$= \frac{1}{2}\log\left(\frac{1}{\det(\Sigma)}\right) + \frac{1}{2}\left((\theta + \theta^*)^\top I(\theta + \theta^*) - \theta^\top \Sigma^{-1}\theta\right). \tag{31}$$

Substituting Eq.31 into Eq.22, we can compute the KL divergence between $Q$ and $P$ as follows. Here, we assume $\Theta = \mathbb{R}^d$.

$$\mathcal{D}(Q\|P)$$

$$= \mathbb{E}_{\theta \sim Q}\left(\frac{Q(\theta)}{P(\theta)}\right)$$

$$= \int_{\theta \in \Theta} \log\left(\frac{q(\theta)}{p(\theta)}\right) q(\theta)\mathrm{d}\theta$$

$$= \int_{\theta \in \Theta} \left[\frac{1}{2}\log\left(\frac{1}{\det(\Sigma)}\right) + \frac{1}{2}\left((\theta + \theta^*)^\top I(\theta + \theta^*) - \theta^\top \Sigma^{-1}\theta\right)\right] q(\theta)\mathrm{d}\theta$$

$$= \frac{1}{2}\log\left(\frac{1}{\det(\Sigma)}\right) + \frac{1}{2}\int_{\theta \in \Theta} \theta^\top I\theta q(\theta)\mathrm{d}\theta + \frac{1}{2}\int_{\theta \in \Theta} \theta^{*\top} I\theta^* q(\theta)\mathrm{d}\theta + \int_{\theta \in \Theta} \theta^{*\top} I\theta q(\theta)\mathrm{d}\theta$$

$$\quad - \frac{1}{2}\int_{\theta \in \Theta} \theta^\top \Sigma^{-1}\theta q(\theta)\mathrm{d}\theta$$

$$= \frac{1}{2}\log\left(\frac{1}{\det(\Sigma)}\right) + \frac{1}{2}\mathbb{E}_{\theta \sim \mathcal{N}(0,\Sigma)}\left[\theta^\top I\theta\right] + \frac{1}{2}\mathbb{E}_{\theta \sim \mathcal{N}(0,\Sigma)}\left[\theta^{*\top} I\theta^*\right] + \mathbb{E}_{\theta \sim \mathcal{N}(0,\Sigma)}\left[\theta^{*\top} I\theta\right]$$

$$\quad - \frac{1}{2}\mathbb{E}_{\theta \sim \mathcal{N}(0,\Sigma)}\left[\theta^\top \Sigma^{-1}\theta\right]$$

$$= \frac{1}{2}\log\left(\frac{1}{\det(\Sigma)}\right) + \frac{1}{2}\mathrm{tr}(\Sigma) + \frac{1}{2}\|\theta^*\|^2 - \frac{1}{2}\mathrm{tr}(I)$$

$$\leq \frac{1}{2}\log\left(\frac{1}{\det(\Sigma)}\right) + \frac{1}{2}\mathrm{tr}(\Sigma) + \frac{1}{2}R^2 - \frac{1}{2}\mathrm{tr}(I). \tag{32}$$

The last transformation is obtained because $\|\theta^*\|^2 \leq R^2$ always holds when the search range of the parameter $\theta$ is in a hypersphere of radius $R$. From Eq.24, we obtain

$$A\Sigma + \Sigma A = \frac{\eta}{|S|}C.$$

Therefore, we obtain

$$A\Sigma A^{-1} + \Sigma = \frac{\eta}{|S|}CA^{-1}. \tag{33}$$

By computing the trace on both sides, the equation becomes

$$\mathrm{tr}\left(A\Sigma A^{-1} + \Sigma\right) = \mathrm{tr}\left(\frac{\eta}{|S|}CA^{-1}\right). \tag{34}$$

The left-hand side (LHS) can be transformed as follows:

$$
\begin{aligned}
\text{LHS} &= \text{tr}\left(A\Sigma A^{-1} + \Sigma\right) \\
&= \text{tr}\left(A\Sigma A^{-1}\right) + \text{tr}\left(\Sigma\right) \\
&= \text{tr}\left(\Sigma A^{-1} A\right) + \text{tr}\left(\Sigma\right) \\
&= \text{tr}\left(\Sigma\right) + \text{tr}\left(\Sigma\right) \\
&= 2\text{tr}\left(\Sigma\right).
\end{aligned}
\tag{35}
$$

Thus, we can derive the following:

$$
\text{tr}\left(\Sigma\right) = \frac{1}{2}\text{tr}\left(\frac{\eta}{|S|}CA^{-1}\right) = \frac{1}{2}\frac{\eta}{|S|}\text{tr}\left(CA^{-1}\right).
\tag{36}
$$

We can also compute the following:

$$
\text{tr}(I) = d,
\tag{37}
$$

because $d$ is the dimension of parameters $\theta$ and $I \in \mathbb{R}^{d \times d}$. Therefore, by substituting Eq.36 and Eq.37 into Eq.32, we obtain

$$
\mathcal{D}(Q\|P) \le \frac{1}{4}\frac{\eta}{|S|}\text{tr}(CA^{-1}) - \frac{1}{2}\log\left(\det(\Sigma)\right) + \frac{1}{2}R^2 - \frac{1}{2}d.
\tag{38}
$$

Eq.38 is an upper bound on the distance between the distribution $Q$ learned by SGD and the a priori distribution $P$. Therefore, by substituting Eq.38 into Eq.21, we can obtain a PAC-Bayesian generalization bound for SGD.

The proof is completed. $\qquad\square$

**Lemma 4** (extension of He et al. (2019), Appendix B.2). *Assuming that $A$ and $\Sigma$ commute, the KL divergence between the distribution $Q$ of SGD and the prior distribution $P$ satisfies the following inequality:*

$$
\mathcal{D}(Q\|P) \le \frac{\eta}{4|S|}\text{tr}(CA^{-1}) + \frac{1}{2}d\log\left(\frac{2|S|}{\eta}\right) - \frac{1}{2}\log\left(\det(CA^{-1})\right) + \frac{1}{2}R^2 - \frac{1}{2}d.
\tag{39}
$$

Lemma 4 provides the distance between the distribution $Q$ obtained by SGD and the prior distribution of the parameters. The assumption that $A$ and $\Sigma$ commute implies that matrices $A$ and $\Sigma$ are simultaneously diagonalizable, and similar assumptions have been used in the work of Jastrzębski et al. (2017) and Liu et al. (2021). Although similar results can be derived without assuming that $A$ and $\Sigma$ commute, we used the above assumption for simplicity. Based on this assumption, we can compute a generalization bound for a constant case.

*Proof of Lemma 4.* Because $A$ and $\Sigma$ commute by assumption, the following equation can be obtained by transforming Eq.24:

$$
\begin{aligned}
A\Sigma + \Sigma A &= \frac{\eta}{|S|}C \\
2\Sigma A &= \frac{\eta}{|S|}C \\
\Sigma &= \frac{\eta}{2|S|}CA^{-1}.
\end{aligned}
\tag{40}
$$

Thus,

$$
\det(\Sigma) = \det\left(\frac{\eta}{2|S|}CA^{-1}\right) = \left(\frac{\eta}{2|S|}\right)^d \det(CA^{-1}),
\tag{41}
$$

and we can compute

$$\log\left(\det(\Sigma)\right) = \log\left[\left(\frac{\eta}{2|S|}\right)^d \det(CA^{-1})\right]$$

$$= -d\log\left(\frac{2|S|}{\eta}\right) + \log\left(\det(CA^{-1})\right). \tag{42}$$

Therefore, Eq.39 can be obtained by substituting Eq.42 into Eq.38.

The proof is completed. □

Thus, we can easily prove Theorem 1.

*Proof of Theorem 1.* It is obtained by substituting Eq.39 into Eq.21.

The proof is completed. □

## B.2 ADDITIONAL TWO FACTORS IN THE PROPOSED GENERALIZATION BOUND

In addition to the three factors discussed in He et al. (2019)'s Theorem 2, the local geometry around minima, gradient fluctuations, and hyperparameters, the generalization bound derived in this study incorporates a factor for the range of parameter search and a factor for network initialization.

**Parameter search space** The norm of $\theta$ after learning appears as the third term $R^2$ in Eq.32. Hence, the parameter search range and weight decay are naturally incorporated into the generalization bound.

**Network initialization.** Moreover, this generalization bound naturally incorporates the network initialization factor. Assuming that the prior distribution is $\mathcal{N}(-\theta^*, \lambda I), 0 < \lambda$, we obtain the following:

$$p'(\theta) = \frac{1}{\sqrt{(2\pi\lambda)^d \det(I)}} \exp\left(-\frac{1}{2\lambda}(\theta + \theta^*)^\top I(\theta + \theta^*)\right), \quad 0 < \lambda. \tag{43}$$

Eq.32 is as follows.

$$\mathcal{D}(Q\|P') \leq \frac{1}{2}\log\left(\frac{\lambda^d}{\det(\Sigma)}\right) + \frac{1}{2\lambda}\mathrm{tr}(\Sigma) + \frac{1}{2\lambda}R^2 - \frac{1}{2}\mathrm{tr}(I). \tag{44}$$

By differentiating the right-hand side (RHS) by $\lambda$, we can observe that RHS is minimized by RHS $= \frac{d}{2}\log\left(\frac{\mathrm{tr}(\Sigma)+\mathrm{R}^2}{d}\right) - \frac{1}{2}\log\left(\det\left(\Sigma\right)\right)$, when $\lambda = \frac{\mathrm{tr}(\Sigma)+\mathrm{R}^2}{d}$. Assuming that $d$ is sufficiently large and $\mathrm{tr}(\Sigma) + \mathrm{R}^2 < \mathrm{d}$ is valid (this property is also called *overparameterization* (Allen-Zhu et al., 2019; Brutzkus et al., 2018; Du et al., 2019)), we can theoretically guarantee that the generalization bound will be tighter when we change the covariance matrix $I$ of the prior distribution to a diagonal matrix with even smaller eigenvalues, such as He's initialization (He et al., 2015) or Xavier's initialization (Glorot & Bengio, 2010). Therefore, the generalization bound proposed in this study, which is an extended version of He et al. (2019), also naturally incorporates factors related to the initialization of the network.

## B.3 GENERALIZATION BOUND FOR THE TIME-VARYING CASE

To prove Theorem 2, we only need to describe how the time-varying batch size and learning rate change Eq.24 in Lemma 2. To do so, we solve following Lemma 5:

**Lemma 5** (extension of Gardiner (2004), pp.115-116). *Under second-order differentiable assumption (Eq.5), if $t$ is sufficiently large and hyperparameter $\frac{\eta}{|S|}$ is time-varying, the Ornstein–Uhlenbeck process (Eq.4)'s distribution*

$$q(\theta) = M \exp\left\{-\frac{1}{2}\theta^\top \Sigma^{-1}\theta\right\} \tag{45}$$

*has the following property:*

$$A\Sigma + \Sigma A = n(\xi)C, \quad \xi \in [0, t]. \tag{46}$$

*Here,*

$$n(t) = \frac{\eta(t)}{|S(t)|}. \tag{47}$$

*Proof of Lemma 5.* From the analytical solution of the Ornstein–Uhlenbeck process (Uhlenbeck & Ornstein, 1930) in the time-varying case (Gardiner, 2004), the parameter $\theta$ can be expressed as follows:

$$\theta(t) = e^{-At}\theta(0) + \int_0^t \sqrt{n(t')}e^{-A(t-t')}B\mathrm{d}W(t'), \tag{48}$$

where $W(t')$ is a white noise and follows $\mathcal{N}(0, I)$. From Eq.45, we obtain

$$\Sigma = \mathbb{E}_{\theta \sim Q}\left[\theta\theta^\top\right]. \tag{49}$$

Thus, we obtain the following equation:

$$
\begin{aligned}
A\Sigma + \Sigma A = {}& Ae^{-At}\langle\theta(0), \theta^\top(0)\rangle e^{-At} + e^{-At}\langle\theta(0), \theta^\top(0)\rangle e^{-At}A \\
& + \int_0^t n(t')Ae^{-A(t-t')}Ce^{-A(t-t')}\mathrm{d}t' \\
& + \int_0^t n(t')e^{-A(t-t')}Ce^{-A(t-t')}A\mathrm{d}t'
\end{aligned}
\tag{50}
$$

where $A$ is the Hessian matrix and symmetric. We use the mean-value theorem of integrals to obtain the following equation: For some $\xi_1, \xi_2 \in [0, t]$,

$$\int_0^t n(t')Ae^{-A(t-t')}Ce^{-A(t-t')}\mathrm{d}t' = n(\xi_1)\int_0^t Ae^{-A(t-t')}Ce^{-A(t-t')}\mathrm{d}t' \tag{51}$$

$$\int_0^t n(t')e^{-A(t-t')}Ce^{-A(t-t')}A\mathrm{d}t' = n(\xi_2)\int_0^t e^{-A(t-t')}Ce^{-A(t-t')}A\mathrm{d}t' \tag{52}$$

Here, we assume that the matrix $D$ is

$$D = \int_0^t e^{-A(t-t')}Ce^{-A(t-t')}\mathrm{d}t' \tag{53}$$

and that $A$ and $D$ commute. Then, because $AD = DA$,

$$\xi = \xi_1 = \xi_2 \tag{54}$$

holds for Eq.51 and Eq.52. Thus, by substituting Eq.54 into Eqs.51 and 52, and then substituting the result into Eq.50, we derive

$$
\begin{aligned}
A\Sigma + \Sigma A = {}& Ae^{-At}\langle\theta(0), \theta^\top(0)\rangle e^{-At} + e^{-At}\langle\theta(0), \theta^\top(0)\rangle e^{-At}A \\
& + n(\xi)\int_0^t Ae^{-A(t-t')}Ce^{-A(t-t')}\mathrm{d}t' \\
& + n(\xi)\int_0^t e^{-A(t-t')}Ce^{-A(t-t')}A\mathrm{d}t' \\
= {}& Ae^{-At}\langle\theta(0), \theta^\top(0)\rangle e^{-At} + e^{-At}\langle\theta(0), \theta^\top(0)\rangle e^{-At}A \\
& + n(\xi)\int_0^t \frac{\mathrm{d}}{\mathrm{d}t'}e^{-A(t-t')}Ce^{-A(t-t')}\mathrm{d}t' \\
= {}& Ae^{-At}\langle\theta(0), \theta^\top(0)\rangle e^{-At} + e^{-At}\langle\theta(0), \theta^\top(0)\rangle e^{-At}A \\
& + n(\xi)C - n(\xi)e^{-At}Ce^{-At} \\
\simeq {}& n(\xi)C.
\end{aligned}
\tag{55}
$$

The last transformation is guided by the assumption that $A$ is a positive definite matrix and all its eigenvalues are positive and by the approximation of the term multiplied by $e^{-At}$ twice with zero, because $t$ is sufficiently large.

The proof is completed. $\qquad\square$

*Proof of Theorem 2.* Using Eq.55, rather than Eq.27 in the proof of Theorem 1, leads to Eq.10. Furthermore, as exactly

$$\min(n(t)) \leq n(\xi) \leq \max(n(t)), \tag{56}$$

by applying a certain $\lambda \in [0,1]$, we can express

$$n(\xi) = (1-\lambda)\min(n(t)) + \lambda\max(n(t)).$$

Therefore, Eq.11 was derived simultaneously.

The proof is completed. $\qquad\square$

### B.4 FUNCTIONAL MODEL OF THE TRAINING ERROR

#### B.4.1 CONSTANT CASE

We derive a training error model for the constant case. We approximate the training error $\hat{\mathcal{R}}(\theta)$ by substituting Eq.7 into Eq.6 and then approximate $\hat{\mathcal{R}}(Q)$ by computing the expected value in the parameter distribution $Q$. In the constant case, we can compute

$$
\begin{aligned}
\hat{\mathcal{R}}(\theta) \simeq \frac{1}{2}(\theta - \theta_b)^\top \hat{A}(\theta - \theta_b) &= \frac{1}{2}\left(e^{-At}\theta(0) - \theta_b + \sqrt{\frac{\eta}{|S|}}\int_0^t e^{-A(t-t')}BdW(t')\right)^\top \\
&\quad \hat{A}\left(e^{-At}\theta(0) - \theta_b + \sqrt{\frac{\eta}{|S|}}\int_0^t e^{-A(t-t')}BdW(t')\right) \\
&= \frac{1}{2}(\theta(0)^\top e^{-At} - \theta_b^\top)\hat{A}(e^{-At}\theta(0) - \theta_b) \\
&\quad + \frac{1}{2}\left(\int_0^t e^{-A(t-t')}BdW(t')\right)^\top \hat{A}\left(\int_0^t e^{-A(t-t')}BdW(t')\right)\frac{\eta}{|S|} \\
&\quad + \left(\left(\int_0^t e^{-A(t-t')}BdW(t')\right)^\top \hat{A}(e^{-At}\theta(0) - \theta_b)\right)\sqrt{\frac{\eta}{|S|}}. \tag{57}
\end{aligned}
$$

Therefore, by taking the expected value with the distribution $Q$, we obtain

$$
\begin{aligned}
\hat{\mathcal{R}}(Q) &= \mathbb{E}_{\theta\sim Q}\left[\hat{\mathcal{R}}(\theta)\right] \\
&\simeq \mathbb{E}_{\theta\sim Q}\left[\frac{1}{2}(\theta - \theta_b)^\top \hat{A}(\theta - \theta_b)\right] \\
&= \mathbb{E}\left[\frac{1}{2}(\theta(0)^\top e^{-At} - \theta_b^\top)\hat{A}(e^{-At}\theta(0) - \theta_b)\right] \\
&\quad + \mathbb{E}\left[\frac{1}{2}\left(\int_0^t e^{-A(t-t')}BdW(t')\right)^\top \hat{A}\left(\int_0^t e^{-A(t-t')}BdW(t')\right)\right]\frac{\eta}{|S|} \\
&\quad + \mathbb{E}\left[\left(\left(\int_0^t e^{-A(t-t')}BdW(t')\right)^\top \hat{A}(e^{-At}\theta(0) - \theta_b)\right)\right]\sqrt{\frac{\eta}{|S|}} \\
&= a_0 + a_1\frac{\eta}{|S|} + a_2\sqrt{\frac{\eta}{|S|}}. \tag{58}
\end{aligned}
$$

This allows us to model $\hat{\mathcal{R}}(Q)$, as shown in Eq.13.

### B.4.2 TIME-VARYING CASE

To compute the training error model for the time-varying case, we only need to consider how the time-varying batch size and learning rate change Eq.6. Thus, we use the mean value theorem of integrals in Eq.9 to obtain the following equation:

$$
\begin{aligned}
\theta(t) &= \theta(0)e^{-At} + \int_0^t \sqrt{n(t')}e^{-A(t-t')}B\mathrm{d}W(t') \\
&= \theta(0)e^{-At} + \sqrt{n(\xi)}\int_0^t e^{-A(t-t')}B\mathrm{d}W(t').
\end{aligned}
\tag{59}
$$

Here, $\xi \in [0, t]$ and as exactly

$$
\min(\sqrt{n(t)}) \le \sqrt{n(t)} \le \max(\sqrt{n(t)}),
\tag{60}
$$

for some $\lambda \in [0, 1]$, we obtain

$$
\sqrt{n(\xi)} = (1 - \lambda)\min(\sqrt{n(t)}) + \lambda\max(\sqrt{n(t)}).
\tag{61}
$$

Thus, we can compute the first term in Eq.16, which is the training error model for the time-varying case, by substituting Eq.61 into Eq.59 and then performing the same transformation as in Appendix B.4.1.

### B.5 FUNCTIONAL MODEL OF THE GENERALIZATION BOUND

### B.5.1 CONSTANT CASE

In this section, under appropriate assumptions, we show that Eq.8 monotonically decreases with respect to $\frac{\eta}{|S|}$ as a basis for deriving Eq.14 based on Eq.8. We recall He et al. (2019)'s proof to make our paper complete. We first make the following assumptions about dimension $d$ of the parameter space $\Theta$.

**Assumption 1** (See He et al. (2019), Assumption 2). *The network size is large enough:*

$$
d > \frac{\mathrm{tr}(CA^{-1})\eta}{2|S|},
\tag{62}
$$

*where $d$ is the number of parameters, $C$ is the magnitude of the individual gradient noise, $A$ is the Hessian matrix around the global minima, $\eta$ is the learning rate, and $|S|$ is the batch size.*

This assumption can be justified by the fact that network sizes of neural networks are often extremely large. This property is also called *overparameterization* (Allen-Zhu et al., 2019; Brutzkus et al., 2018; Du et al., 2019). We can obtain the following corollary by combining Eq.8 and Assumption 1.

**Corollary 1** (See He et al. (2019), Appendix B.3). *When all conditions of Theorem 1 and Assumption 1 hold, the generalization bound of the network is negatively correlated with the ratio of the learning rate to the batch size.*

*Proof of Corollary 1.* We first define

$$
I = \frac{\eta}{2|S|}\mathrm{tr}(CA^{-1}) + d\log\left(\frac{2|S|}{\eta}\right) - \log\left(\det\left(CA^{-1}\right)\right) + R^2 - d + 2\log\left(\frac{1}{\delta}\right) + 2\log N + 4.
\tag{63}
$$

Then, Eq.8 becomes

$$
\mathcal{R}(Q) \le \hat{\mathcal{R}}(Q) + \sqrt{\frac{I}{4N - 2}}.
\tag{64}
$$

We calculate the derivative of $I$ with respect to the ratio $\frac{\eta}{|S|}$ to check whether the generalization bound has a negative correlation with the ratio. For brevity, we set $k = \frac{\eta}{|S|}$:

$$
\begin{aligned}
\frac{\partial I}{\partial k} &= \frac{\partial}{\partial k}\left[\frac{k}{2}\mathrm{tr}(CA^{-1}) - d\log(2k) - \log\left(\det\left(CA^{-1}\right)\right) + R^2 - d + 2\log\left(\frac{1}{\delta}\right) + 2\log N + 4\right] \\
&= \frac{\mathrm{tr}(CA^{-1})}{2} - \frac{d}{k}.
\end{aligned}
\tag{65}
$$

Therefore, when Assumption 1 holds, we have

$$
d > \frac{\mathrm{tr}(CA^{-1})\eta}{2|S|} = \frac{k}{2}\mathrm{tr}(CA^{-1}).
\tag{66}
$$

Thus,

$$
\frac{\partial I}{\partial k} < 0.
\tag{67}
$$

Then, $I$ and the generalization bound have a negative correlation with the ratio of the learning rate to the batch size.

The proof is competed. $\qquad\square$

### B.5.2 TIME-VARYING CASE

In this section, the basis for deriving the second term in Eq.16 based on Eq.10, we will show that Eq.10 monotonically decreases with respect to $n_2$ in Eq.18 under appropriate assumptions. First, we make the following assumptions about the dimension $d$ of the parameter space $\Theta$.

**Assumption 2** (cf. He et al. (2019), Assumption 2). *The network size is large enough:*

$$
d > \frac{\mathrm{tr}(CA^{-1})}{2}n_2,
\tag{68}
$$

*where $d$ is the number of parameters, $C$ is the magnitude of the individual gradient noise, $A$ is the Hessian matrix around the global minima, and $n_2$ is the intermediate value of $\frac{\eta(t)}{|S(t)|}$ in Eq.18, $\eta(t)$ is the learning rate, and $|S(t)|$ is the batch size.*

This assumption can also be justified as Assumption 1 when the network sizes of neural networks are often extremely large. We can obtain the following corollary by combining Eq.10 and Assumption 2.

**Corollary 2** (extension of He et al. (2019), Appendix B.3). *When all conditions of Theorem 2 and Assumption 2 hold, the generalization bound of the network is negatively correlated with the ratio of the learning rate to the batch size.*

*Proof of Corollary 2.* Along the proof of Corollary 1, we use Eq.10, $k = n_2$ and Assumption 2, rather than Eq.8, $k = \frac{\eta}{|S|}$ and Assumption 1. Then, we can obtain

$$
\frac{\partial I}{\partial k} < 0.
\tag{69}
$$

The proof is competed. $\qquad\square$

## C    IMPLEMENTATION DETAIL

### C.1    DATASETS AND MODELS

#### C.1.1    DATASETS

We tested the proposed model on two popular image datasets. CIFAR10 and CIFAR100 (Krizhevsky et al., 2009): 60 K natural RGB images of 10 classes for CIFAR10 and 100 classes for CIFAR100 with a train/test split of 50K/10 K. For both datasets, we use PyTorch version[1].

#### C.1.2    MODELS

We tested the proposed model using two popular model networks. For CIFAR10, we used VGG16 (Simonyan & Zisserman, 2015), and for CIFAR100, we use WRN28-10 (Zagoruyko & Komodakis, 2016). For both model networks, we build on the code from the implementation of Kim (2018).

#### C.1.3    TRAINING

We modified the implementation of Kim (2018). Both datasets are normalized by mean and standard deviation. In the main experiments, training was performed via SGD with a momentum of 0.9, and weight decay of 5e-4 for 200 epochs. We began training with a learning rate of $\eta_{init}$, run for 200 epochs, and reduced by a multiplicative factor of $\gamma$ after 60, 120, and 160 epochs to make $\eta_{final} = \gamma^3 \eta_{init}$.

### C.2    ERROR ESTIMATION EXPERIMENT

#### C.2.1    EXPERIMENTAL DETAILS

As described in Section 6.2, we fit the parameters of $\phi$ to minimize $\delta(S, \eta; \phi)$, using least squares regression. In doing so, we used `scipy.optimize.curve_fit`[2]. The optimal parameter $\phi^*$ is given by

$$\phi^* = \arg\min_{\phi} \sum_{S,\eta} |\delta(S, \eta; \phi)|^2 . \tag{70}$$

To measure the performance of the proposed model, the mean $\mu$ and standard deviation $\sigma$ of the relative error $\delta(S, \eta; \phi^*)$ are computed based on the fitted parameter $\phi^*$. To evaluate the stability of the model, we fit a parameter $\phi$ based on a randomly sampled generalization error $\epsilon$ and compute $\mu$ and $\sigma$ from the relative error $\delta(S, \eta; \phi)$. The values of $\mu$ and $\sigma$ obtained after 100 repetitions are shaded within one standard deviation, as shown in Figure.2.

#### C.2.2    FOUND PHI VALUES

**Table 1:** The optimal value $\phi^*$ is determined by the least squares regression of relative error.

**(a)** Constant case

|          | $c_0$  | $c_1$                 | $c_2$              | $c_3$                 | $c_4$ |
|----------|--------|-----------------------|--------------------|-----------------------|-------|
| CIFAR10  | 81.92  | $-7.87 \cdot 10^{-1}$ | $1.00 \cdot 10^{-4}$ | $8.20 \cdot 10^{-2}$  | 301   |
| CIFAR100 | 8.97   | 6.45                  | $7.66 \cdot 10^{-4}$ | $1.50 \cdot 10^{-1}$  | 7.69  |

**(b)** Time-varying case

|          | $c_0$  | $c_1$ | $c_2$                | $c_3$                | $c_4$ | $\lambda_1$          | $\lambda_2$          |
|----------|--------|-------|----------------------|----------------------|-------|----------------------|----------------------|
| CIFAR10  | 57.68  | 1.17  | $1.56 \cdot 10^{-4}$ | $5.23 \cdot 10^{-2}$ | 709   | $8.35 \cdot 10^{-1}$ | $8.89 \cdot 10^{-2}$ |
| CIFAR100 | 34.77  | 6.91  | $9.52 \cdot 10^{-4}$ | $1.21 \cdot 10^{-1}$ | 4.03  | $6.00 \cdot 10^{-1}$ | $1.63 \cdot 10^{-1}$ |

---

[1]`https://github.com/pytorch/vision`
[2]`https://github.com/scipy/scipy`

### C.2.3 COMPARISON MODELS FOR THE TIME-VARYING CASE

For the time-varying case in this study, we used a step decay for the learning rate. Thus, rather than using the mean value theorem of integrals to make approximations, as in Eqs.55 and59, we can obtain a more accurate model by treating the integral value as a constant for each interval as follows:

$$\int_0^t n(t')F(t')\mathrm{d}t' = n(0)\int_0^{t_0} F(t')\mathrm{d}t' + n(t_0)\int_{t_0}^t F(t')\mathrm{d}t' = n(0)I_0 + n(t_0)I_1. \quad (71)$$

We call this the analytical solution model. We call the model using only the final learning rate as the stationary model and fit these two models to the generalization error. The comparison results can be found in Table.2.

**Table 2:** Fitting results of the proposed model and the comparison models for the time-varying case.

**(a) CIFAR10**

|  | $|\phi|$ | $\mu$ (% ↓) | $\sigma$ (% ↓) | AIC (↓) |
|---|---|---|---|---|
| stationary model | 5 | -6.03 | 23.89 | 5.08 |
| analytical solution model | 13 | **-2.66** | **16.15** | -98.79 |
| proposed model | 7 | -2.71 | 16.28 | **-108.21** |

**(b) CIFAR100**

|  | $|\phi|$ | $\mu$ (% ↓) | $\sigma$ (% ↓) | AIC (↓) |
|---|---|---|---|---|
| stationary model | 5 | -4.88 | 21.88 | -27.97 |
| analytical solution model | 13 | **-0.86** | **9.57** | -316.23 |
| proposed model | 7 | -0.88 | 9.69 | **-323.80** |

Here, the analytical solution model uses step decay and can divide the integration interval into four, and then $|\phi| = 13$. Meanwhile, the stationary model has the same number of parameters as the constant case, $|\phi| = 5$. As shown in these results, the analytical solution model obtains the smallest relative error, and the proposed model has almost the same performance with a smaller number of parameters. Assuming that the relative error $\delta$ follows a normal distribution $\mathcal{N}(\mu, \sigma)$ for the mean $\mu$ and standard deviation $\sigma$ of the relative error $\delta$ in the fitted model, we can observe that Akaike's information criterion (AIC) (Akaike, 1998) is minimized by the proposed model. Thus, the integral in the proposed model can be approximated sufficiently using the mean value theorem.

## C.3 STABILITY EXPERIMENT

### C.3.1 EXPERIMENTAL DETAILS

We use the same experimental setup as in Appendix C.2.1. However, there is instability in `scipy.optimize.curve_fit` with respect to the initial value of the search parameter $\phi$. Thus, we sampled an initial value of $\phi$ from a uniform distribution $[0, 1]$ and fitted it several times (10 and 5 times for the constant and time-varying cases, respectively). Then, we used the parameter that best fit the sampled data for the stability experiment.

## C.4 HYPERPARAMETER OPTIMIZATION EXPERIMENT

### C.4.1 BAYESIAN OPTIMIZATION MODEL FOR THE CONSTANT CASE

In this section, we describe the implementation details of the original kernel function proposed in Eq.20. To construct the proposed kernel function, we implemented the following kernel function using `sklearn.gaussian_process.kernels`[3]:

```python
from sklearn.gaussian_process.kernels import DotProduct, ConstantKernel

sigma_0_bounds = (1e-10, 1e10)
equation:proposed_kernel_constant_case = (DotProduct(sigma_0_bounds=sigma_0_bounds) ** 0.5
                + DotProduct(sigma_0_bounds=sigma_0_bounds)
                + DotProduct(sigma_0_bounds=sigma_0_bounds) ** -0.5
                + ConstantKernel())
```

`sigma_0_bounds` decides the range of the $\sigma$, which controls the inhomogenity of the kernel.

### C.4.2 ALGORITHM FOR THE TIME-VARYING CASE

The proposed model Eq.16 for the time-varying case cannot be represented by a linear combination of $y = \boldsymbol{w}^\top \boldsymbol{\psi}(x)$, as the product term of the parameters of $c, \lambda$ appears. Thus, we employed a hyperparameter search by sequential model-based global optimization (Bergstra et al., 2011) combining the Levenberg–Marquardt algorithm (LM) (Moré, 1978) and Thompson Sampling (Thompson, 1933; Russo et al., 2017), directly using the proposed model. The LM is used to solve non-linear least squares problems. It interpolates between the Gauss—Newton algorithm (GNA) and the GD method. It is more robust than the GNA, which implies that, in many cases, it finds a solution even if it starts very far from the final minimum. It can also be regarded as the GNA using a trust region approach. The distribution of the parameter $\phi$ is obtained by minimizing the relative error of $\delta(S, \eta; \boldsymbol{\phi}) = \frac{\hat{\epsilon}(S,\eta;\phi) - \epsilon(S,\eta)}{\epsilon(S,\eta)}$ with respect to the parameter $\phi$ by the LM. The algorithm is as follows.

---

**Algorithm 1** Hyperparameter search algorithm for the time-varying case

---

$\mathcal{H} \leftarrow \varnothing$
sample $\boldsymbol{\phi}_0$ from $U(0, 1)$
**for** $t \leftarrow 1$ to $T$ **do**
$\quad \frac{\eta}{|S|}^*_{max}, \frac{\eta}{|S|}^*_{min} \leftarrow \arg\min_{\frac{\eta}{|S|}_{max}, \frac{\eta}{|S|}_{min}} \delta(S, \eta; \boldsymbol{\phi}_{t-1})$
$\quad$ Evaluate $f(\frac{\eta}{|S|}^*_{max}, \frac{\eta}{|S|}^*_{min})$ $\qquad\qquad\qquad\qquad\qquad$ ▷ Expensive step
$\quad \mathcal{H} \leftarrow \mathcal{H} \cup ((\frac{\eta}{|S|}^*_{max}, \frac{\eta}{|S|}^*_{min}), f(\frac{\eta}{|S|}^*_{max}, \frac{\eta}{|S|}^*_{min}))$
$\quad \boldsymbol{\mu}, \boldsymbol{\Sigma} \leftarrow \text{LM}(\delta, \mathcal{H})$
$\quad$ sample $\boldsymbol{\phi}_t$ from trunc-norm$(\boldsymbol{\mu}, \sqrt{\text{diag}(\boldsymbol{\Sigma})}, \boldsymbol{b}_{lower}, \boldsymbol{b}_{upper})$
**end for**
**return** $\mathcal{H}$

---

Here, $\mathcal{H}$ represents the history of observations, and $f(\frac{\eta}{|S|}^*_{max}, \frac{\eta}{|S|}^*_{min})$ represents the evaluation of the network for a hyperparameter $(\frac{\eta}{|S|}^*_{max}, \frac{\eta}{|S|}^*_{min})$. To keep the parameter $\phi$ in this range, Thompson sampling is performed for each parameter from the following truncated normal distribution using the lower bound $\boldsymbol{b}_{lower}$ and upper bound $\boldsymbol{b}_{upper}$.

$$f(x; \mu, \sigma, b_{lower}, b_{upper}) = \frac{1}{\sigma} \frac{\varphi(\frac{x-\mu}{\sigma})}{\Phi(\frac{b_{upper}-\mu}{\sigma}) - \Phi(\frac{b_{lower}-\mu}{\sigma})}, \tag{72}$$

where $\varphi(x)$ is the probability density function of the standard normal distribution as follows:

$$\varphi(x) = \frac{1}{\sqrt{2\pi}} \exp\left(-\frac{1}{2}x^2\right), \tag{73}$$

---

[3]https://github.com/scikit-learn/scikit-learn

and $\Phi(x)$ is its cumulative distribution function:

$$\Phi(x) = \frac{1}{2}\left(1 + \text{erf}\left(\frac{x}{\sqrt{2}}\right)\right).$$ (74)

### C.4.3 EXPERIMENTAL DETAILS

In this section, we present the implementation details of the hyperparameter optimization experiments for comparison. In both constant and time-varying cases, $\frac{\eta}{|S|}$ was allowed to explore in the range of $[10^{-7}, 10^{-2}]$ for CIFAR10 and in the range of $[10^{-6}, 10^{-2}]$ for CIFAR100. These were determined based on the experimental results for each dataset (Figures.1a and 1b). We computed the maximum test accuracy obtained over 30 rounds of hyperparameter search. By performing this for a total of 100 times, we computed a 95% confidence interval of the maximum accuracy. The proposed method uses a uniform distribution search, whereas the comparison method uses a uniform distribution search and log-uniform search. For comparison, we used Bayesian optimization[4] (Nogueira, 2014) using the Matern kernel and RBF kernel, Hyperopt[5] (Bergstra et al., 2015), and Optuna[6] (Akiba et al., 2019).

---

[4]https://github.com/fmfn/BayesianOptimization
[5]https://github.com/hyperopt/hyperopt
[6]https://github.com/optuna/optuna

