# OpenReview forum: "A Theoretical and Empirical Model of the Generalization Error under Time-Varying Learning Rate"
_ICLR.cc/2022/Conference — ICLR 2022 Submitted_

### Official Review · Reviewer_8e8R · 2021-10-28

**Correctness:** 2
**Technical Novelty And Significance:** 2
**Empirical Novelty And Significance:** 2
**Recommendation:** 3
**Confidence:** 3

**Main Review:**


-> My main concern is that I am afraid the authors analysis only applies to convex optimization. The main reason is that the eigenvalues of the Hessian at the quadratic minimum also depend in the hyperparameter like learning rate (see for example https://arxiv.org/abs/2003.02218).

-> While one might be able to fit the generalization error to this formula, it does not mean that this formula was actually use by the model, because without a no microscopic understanding of the coefficients.

-> The generalization bound seems to depend only on min and max learning rates. This seems rather weak for hyperparameter tuning because there are many possible learning rate functions with same min and max lr and different generalization. Furthermore, the dependence on the batch size is only through eta/S and it has been shown that generalization is more nuanced (see for example https://arxiv.org/abs/1811.03600).

-> For the hyperparameter tuning comparison to be meaningful, I would recommend comparing with a grid search over parameters. Also, as one changes the learning rate/ batch_size the time to convergence might vary.

-> Some experiments use momentum while the theory uses SGD.

**Summary Of The Paper:**

The authors study the learning rate over batch size dependence of generalization bounds and try to use them for hyperparameter tuning.

In section 4, the authors study generalization bounds for convex optimization with the motivation that this helps understand neural networks. In section 5, the authors use the bounds to propose a functional approximation for the generalization based as a function of the n=eta/|S| scaling of the fluctuations around the minimum, including for the possibility that n is time dependent. The generalization error is roughly a sum of integer powers of sqrt(n). In section 6, the authors the authors fit a set of points to the generalization formula they found. In section 7, the authors use this expression to look for optimal hyperparameters and compare them with other hyperparameter tuning methods.

**Summary Of The Review:**

The simplifications of the paper are too big to make any of the claims relevant for real networks.

Main concern is that their functional form only applies to linear regression with SGD.

 The functional form that they obtain has very limited predictive power and can be seen as a taylor approximation in sqrt{eta/B}.

---

> ### Author Response · Authors · 2021-11-17
> **Response to Reviewer 8e8R**
>
> We would like to thank you for your valuable comments. We now address all your questions and comments as follows:
>
> > My main concern is that I am afraid the authors analysis only applies to convex optimization. The main reason is that the eigenvalues of the Hessian at the quadratic minimum also depend in the hyperparameter like learning rate (see for example https://arxiv.org/abs/2003.02218).
>
> Thank you for your valuable remarks. As a rationale that this research can be used for more than convex optimization, we implicitly assume that after several iterations of SGD, the parameters finally converge close to an attractor local minimum θ* with hessian A. Also, as presented in related work (https://arxiv.org/abs/1712.09913), training of networks with sophisticated architectures (especially ResNet) can be approximated to convex optimization. As you said, the hessian A depends on the time variation of the learning rate (the choice of minima in SGD is discussed in e.g. https://arxiv.org/abs/2002.03495). However, in this study, we consider it as a constant matrix for modeling purposes. In our experiments, we have found that this works well, but of course it is not a perfect modeling. I would like to make it my future work to incorporate the findings of the paper you mentioned (https://arxiv.org/abs/2003.02218), and model the selection of the minima (= Hessian A) in SGD more explicitly. Thank you very much.
>
> > While one might be able to fit the generalization error to this formula, it does not mean that this formula was actually use by the model, because without a no microscopic understanding of the coefficients.
>
> Thank you very much for your valuable suggestions. We will try to further loosen the simplifications and assumptions to make the model more theoretically guaranteed.
>
> > The generalization bound seems to depend only on min and max learning rates. This seems rather weak for hyperparameter tuning because there are many possible learning rate functions with same min and max lr and different generalization. Furthermore, the dependence on the batch size is only through eta/S and it has been shown that generalization is more nuanced (see for example  https://arxiv.org/abs/1811.03600).
>
> Thank you very much for your valuable suggestions. As you said, in this study, we have characterized the temporal variation of the hyperparameters only by only the maximum and minimum values, but we know that this is not enough. However, on the other hand, we would like to draw your attention to the fact that the experimental modeling in this study has been successful. We are considering more detailed modeling of the time variation of the hyperparameters as future work. Also, we simply treat the batch size in this study because we only consider the case where the batch size is small enough compared to the size of the dataset. However, we have not considered the case where the batch size is too large, so we will work on this part as future work. Thank you very much.
>
> > For the hyperparameter tuning comparison to be meaningful, I would recommend comparing with a grid search over parameters. Also, as one changes the learning rate/ batch_size the time to convergence might vary.
>
> Thank you for your valuable remarks. In general, an adaptive algorithm is better than Grid search, which is a non-adaptive algorithm, so I do not think there is a need for comparison. This study does not take convergence time into account, so it is a comparison of hyperpararameter tuning at the algorithmic level. We believe that the comparison at the algorithm level is meaningful enough, but we will also work on the comparison at the application level, which is a more practical case.
>
> > Some experiments use momentum while the theory uses SGD.
>
> Thank you very much for your valuable remarks. This study shows the modeling for SGD, but I think you can see from the related work (https://arxiv.org/abs/1704. 04289) that momentum SGD can be modeled in exactly the same way as for SGD, as long as the momentum parameter is constant. I would like to work on the modeling including Momentum SGD as a future work. Thank you very much.

---

### Official Review · Reviewer_P32Z · 2021-11-01

**Correctness:** 2
**Technical Novelty And Significance:** 3
**Empirical Novelty And Significance:** 3
**Recommendation:** 5
**Confidence:** 4

**Main Review:**

The primary strength of the paper lies in the simplicity of the strategy used in deriving the functional form. The extensive study involving overparametrized neural networks and large-scale vision datasets to show that the functional form approximates the generalization error is quite interesting. The authors also showcase the strength of their proposed model in hyperparameter optimization.


Questions:

1) The covariance matrix $C$ being assumed a constant for all $\theta$ is a very strong assumption, which is very unlikely to hold true for the overparametrized neural networks in practice. Can the authors somehow loosen the assumption by using some form of average covariance matrix over the trajectory?

2) PP 109-110 in Gardiner, 2004 shows the analytic solution to eq. (4) when $g(\theta) = A \theta$. Are the authors assuming the risk as $\frac{1}{2} \theta^T A \theta$ throughout the trajectory starting from some random initialization $\theta(0)$? That will imply, the authors are only considering a quadratic loss function, which has a single global minimum.

One way to handle the issues mentioned in (1) and (2) can be the following:
Assume that after several iterations of SGD, the parameters finally converge close to an attractor local minimum $\theta^{\star}$ with hessian $A$. Then, with $\theta(0)$ starting in a neighborhood around $\theta^{\star}$, assuming that the covariance matrix $C$ is nearly a constant in the neighborhood, the authors can get eq. (7) as the analytic solution of the final distribution around the local minimum.


3) [Experiments] How were the models initialized in the 60 VGG experiments? Was the initialization the same in all the cases? For each LR / batch size ratio, how does the generalization change across multiple initializations? Does the fitted curve change drastically when one considers multiple initializations for different LR / batch size ratios?



**Summary Of The Paper:**

The authors study the dependence of generalization error of trained neural networks on the batch size and the learning rate used in SGD. The primary motivation is to extend the previous works to the case where batch size and learning rate change over the trajectory. The authors present a functional form using PAC-based generalization bounds to model the desired dependence. They conduct extensive experimentation to show that the functional form approximates the generalization error well. In addition, the authors show that a hyperparameter search based on the proposed model outperforms existing hyperparameter optimization libraries like Hyperopt and Optuna.

**Summary Of The Review:**

Overall, I believe the primary strength of the paper lies in the extensive experimental study to corroborate their claim. The major theorems need some restructuring to remove the confusions that may arise (e.g., see (1) and (2) in my review) from the current state. Hence, my scores are currently on the borderline. I am happy to increase my scores after discussing with the authors and other reviewers in the discussion period.

---

> ### Author Response · Authors · 2021-11-17
> **Response to Reviewer P32Z**
>
> We would like to thank you for your valuable comments. We now address all your questions and comments as follows:
>
> > 1. The covariance matrix C being assumed a constant for all θ is a very strong assumption, which is very unlikely to hold true for the overparametrized neural networks in practice. Can the authors somehow loosen the assumption by using some form of average covariance matrix over the trajectory?
> >
> > 2. PP 109-110 in Gardiner, 2004 shows the analytic solution to eq. (4) when g(θ)=Aθ. Are the authors assuming the risk as 12θTAθ throughout the trajectory starting from some random initialization θ(0)? That will imply, the authors are only considering a quadratic loss function, which has a single global minimum.
>
> Thank you for your valuable questions and suggestions. As you said, we implicitly assume that after several iterations of SGD, the parameters finally converge close to an attractor local minimum θ* with hessian A. However, the choice of minima in SGD has been discussed in (e.g. https://arxiv.org/abs/2002.03495), and I would like to incorporate these findings into a more explicit modeling. Thank you very much.
>
> > 3. [Experiments] How were the models initialized in the 60 VGG experiments? Was the initialization the same in all the cases? For each LR / batch size ratio, how does the generalization change across multiple initializations? Does the fitted curve change drastically when one considers multiple initializations for different LR / batch size ratios?
>
> Thank you for your valuable question. In this research, when we train VGG and WRN, we do not specifically have the same initial values (we do not fix the seed values, etc.). Although the training results of the networks change for different seed values, we believe that the fitted curve will not change significantly because it can be approximated to a normal distribution (see e.g. https://arxiv.org/abs/2109.08203v1).

---

### Official Review · Reviewer_GHoM · 2021-11-01

**Correctness:** 2
**Technical Novelty And Significance:** 2
**Empirical Novelty And Significance:** 2
**Recommendation:** 3
**Confidence:** 4

**Main Review:**

This looked on the face of it like a strong theoretical paper building on some interesting work to solve a hard problem.  The originality seems to be using PAC-Bayes bounds to model the generalisation performance and to extend previous work by He et al. (2019) to time varying parameters.

Unfortunately, the analysis was not very convincing on both of these aspects.  Having obtained a PAC-Bayes bounds the authors show under a reasonable assumption that it is negatively correlated with the ratio of the learning rate to the mini-batch size (a ratio they call k).  They then propose to model this by equation (14).  Given the infinity of functions that are negatively correlated to choose this function with no justification (it certainly didn't come from their PAC-Bayes bound) is puzzling and rather suggests that the PAC-Bayes bound is not really used.  On the time vary-case the authors don't seem to have noticed that in solving the Stochastic Differential Equation they have scaled the time with the learning rate.  Thus technically equation (9) is wrong (A is a function of time and their should be an integral in the exponent), but there whole analysis doesn't make much sense because they haven't really capture the dynamics.  The model completely ignores the fact that the time changes with the learning rate (which is very important when using a fixed number of epochs).  There are a number of other strange assumptions in their model.  For example, Equation (7) suggests that for sufficiently long time and small learning rate \theta would go to 0, thus minimising the true risk and not the empirical risk.  This is clearly wrong.  Because you have a limited dataset your gradients of  mini-batch have a bias that means they will minimise the empirical risk, Equation (6).

The reason the authors get away with half models that don't make much sense is because ultimately they are just fitting rather simple empirical data to some function form. (e.g. Equations (15) and (16)).  This is not really a test of any of the analysis being carried out.  The authors start out with what looks like a reasonably principled approach, but then make unjustified assumption and uncontrolled approximations to obtain what they call the functional form.  Another example of this is the use of the mean-valued theorem.  This provides a bound, but that bound could dramatically change the function form.  That is, you can't assume that \lambda does not depend on n(t).

I am not sure whether this paper really tells us much.  It contains a lot of algebra, but on close examination none of its seems to hold up.  Of course, the authors are trying to solve a very hard question and they should be commended for this, but they need to show that their modelling assumptions make sense.  Throwing away almost everything to come up with some a functional form really isn't informative.

**Summary Of The Paper:**

This paper approximates the learning dynamics by an Ornstein-Uhlenbeck process and uses some PAC-Bayes results to obtain a functional form for the generalisation behaviour as a function of the ration of the learning rate to the batch size.  It obtains results for both constant and time-varying parameters.  It then uses these to develop a kernel function for hyper-parameter optimisation using Gaussian Processes modelling.

**Summary Of The Review:**

Superficially this paper looks like a rigorous analysis, but it makes so many dubious assumptions and approximations to ultimately obtaining some rather unconvincing functional forms with enough free parameters to fit almost anything.  This kind of analysis requires far more care, attention to detail and empirical validation than the current paper provides.

---

> ### Author Response · Authors · 2021-11-17
> **Response to Reviewer GHoM**
>
> We would like to thank you for your valuable comments. We now address all your questions and comments as follows:
>
> > Having obtained a PAC-Bayes bounds the authors show under a reasonable assumption that it is negatively correlated with the ratio of the learning rate to the mini-batch size (a ratio they call k). They then propose to model this by equation (14). Given the infinity of functions that are negatively correlated to choose this function with no justification (it certainly didn't come from their PAC-Bayes bound) is puzzling and rather suggests that the PAC-Bayes bound is not really used.
>
> Thank you for your valuable remarks. As you said, this research includes the problem that PAC-Bayes bound is difficult to explain the actual generalization bounds. Therefore, PAC-Bayes bound is only an aid to understand the actual generalization bound, and the generalization bound model of the proposed method has to be heuristic. In fact, the model proposed in Equation (14) is based on the experimental results of related work (https://papers.nips.cc/paper/2019/hash/dc6a70712a252123c40d2adba6a11d84-Abstract.html). The validity of the generalized bound model may be obtained to some extent by searching for a functional form in a wider range (e.g., the exponential part is also considered as a parameter). As you pointed out, the generalized bound model of the proposed method is not completely guaranteed theoretically. But on the other hand, we think it is worth noting that the model predicts the generalization error well through experiments. One of our important future works will be to come up with a generalized bound model with more theoretical guarantees.
>
> > On the time vary-case the authors don't seem to have noticed that in solving the Stochastic Differential Equation they have scaled the time with the learning rate. Thus technically equation (9) is wrong (A is a function of time and their should be an integral in the exponent), but there whole analysis doesn't make much sense because they haven't really capture the dynamics. The model completely ignores the fact that the time changes with the learning rate (which is very important when using a fixed number of epochs).
>
> Thank you for your valuable remarks. You are right that the part about A in equation (9) should also be time-varying. In this study, we have simplified it for modeling purposes, but we believe that this part can be ignored when the learning rate is small enough and the number of epochs is large enough to model the generalization error in practice (I think you will find that the proposed model fits well experimentally.)
>
> > There are a number of other strange assumptions in their model. For example, Equation (7) suggests that for sufficiently long time and small learning rate \theta would go to 0, thus minimising the true risk and not the empirical risk. This is clearly wrong. Because you have a limited dataset your gradients of mini-batch have a bias that means they will minimise the empirical risk, Equation (6).
>
> Thank you for your valuable remarks. In this study, we optimize based on equation (7) for simple modeling, and the contradiction you pointed out is mentioned in section 5.1. We model the training error and the generalization bound separately, so that even when $\theta$ goes to zero, the generalization bound is actually larger, which solves this inconsistency. However, in order to loosen the simplification in this study and get a more accurate model, I will also consider optimization based on equation (6). Thank you very much.
>
> > The reason the authors get away with half models that don't make much sense is because ultimately they are just fitting rather simple empirical data to some function form. (e.g. Equations (15) and (16)). This is not really a test of any of the analysis being carried out. The authors start out with what looks like a reasonably principled approach, but then make unjustified assumption and uncontrolled approximations to obtain what they call the functional form. Another example of this is the use of the mean-valued theorem. This provides a bound, but that bound could dramatically change the function form. That is, you can't assume that \lambda does not depend on n(t).
>
> Thank you very much for your valuable comments. We have already conducted more than 300 experiments for CIFAR10,100, and as future work, we would like to empirically show that the proposed model is correct by increasing the experimental data. As for the approximation using the mean-value theorem, while it has been experimentally shown to work well in this study, as you pointed out, it is a very simple approximation and does not perfectly capture the time-varying case. However, I would like to draw your attention to the fact that this is a completely new study that models the generalization error for batch size and learning rate, and also tries to capture the time-varying learning rate and batch size.

---

### Official Review · Reviewer_sp7W · 2021-11-03

**Correctness:** 4
**Technical Novelty And Significance:** 3
**Empirical Novelty And Significance:** 3
**Recommendation:** 5
**Confidence:** 4

**Main Review:**

Strengths:
* Derives novel generalization bound for time-varying (learning rate / batch size) ratio.
* Proposes a novel functional form for the generalization error w.r.t. the (learning rate / batch size) ratio, and empirically show that it is empirically realistic for image classification tasks.
* Proposes a novel model-based hyperparameter optimization method, which outperforms Bayes optimization baselines that search over the uniform range.

Weaknesses:
* The proposed model-based hyperparameter optimization searches in uniform range --- I find the argument for using this rather than the logarithmic range baselines not convincing. I don’t see how searching in the uniform range is **more theoretically sound** than the logarithmic range. As for practical usefulness, the proposed method can only match the logarithmic-range baselines in final accuracy with slower convergence.

Questions:
* To approximate the learning dynamics as an SDE, the learning rate needs to be small. However, in the experiments, the learning rates aren’t necessarily always small throughout training (e.g. 1/(2^i) i=2,..., 12). Also, larger learning rate corresponds to poorly fitted parts of the models (figure 1). This seems to suggest the proposed hyperparameter optimization method won’t work very well for larger learning rate. Does that agree with your experiment results, and any ideas on how to solve this issue?
* The model treats terms that contain t as constants (e.g. a0, a1, …, a4). This seems to assume the total number of training epochs to be fixed. However, in practice, early stopping is often needed for better generalization. How would the proposed method deal with potential early stopping?


**Summary Of The Paper:**

The paper considers the functional relationship between (time-varying) learning rate & batch size and the generalization error. The contributions are summarized below:
* Assuming quadratic risk function and the SDE limit, the authors derive a novel theoretical generalization bound when the (learning rate / batch size) ratio monotonically decreases.
* Inspired by the generalization bound, the authors propose a heuristic functional form of the generalization error w.r.t. the (learning rate / batch size) ratio.
* The authors conduct experiments on CIFAR10 and CIFAR100 datasets, and show that the heuristic generalization error model can fit the actual generalization curve relatively well.
* The authors propose a novel kernel function for Bayesian optimization in hyperparameter optimization. The kernel function is inspired by the heuristic generalization error model. Empirical results show that the proposed kernel can match or outperform existing Bayes optimization baselines.


**Summary Of The Review:**

For now, I recommend weak rejection.

I think it is very interesting work that yields novel insights. However the main concerns I have is the limited scope and practical usefulness of the proposed theory / method (seems that it’s limited to a fixed # of epochs and performance isn’t better compared to log-scale Bayes opt methods). I will consider raising my score if my questions are satisfyingly addressed.

---

> ### Author Response · Authors · 2021-11-17
> **Response to Reviewer sp7W**
>
> We would like to thank you for your valuable comments. We now address all your questions and comments as follows:
>
> > The proposed model-based hyperparameter optimization searches in uniform range --- I find the argument for using this rather than the logarithmic range baselines not convincing. I don’t see how searching in the uniform range is more theoretically sound than the logarithmic range. As for practical usefulness, the proposed method can only match the logarithmic-range baselines in final accuracy with slower convergence.
>
> Thank you for your valuable remarks. As you said, the proposed method has the same performance as the baseline in the logarithmic range but converges slower than it. We cite the fact that HyperOpt does not perform well in the logarithmic range as a reason why it is not theoretically guaranteed to search in the logarithmic range. However, I realized that this is certainly not enough to say that the uniform range search is theoretically sound. Therefore, we would like to revise our argument to say that the baseline in the logarithmic range is also equivalent in theoretical soundness. However, we still believe that the proposed method has novelty in the search for the uniform range.
>
> > To approximate the learning dynamics as an SDE, the learning rate needs to be small. However, in the experiments, the learning rates aren’t necessarily always small throughout training (e.g. 1/(2^i) i=2,..., 12). Also, larger learning rate corresponds to poorly fitted parts of the models (figure 1). This seems to suggest the proposed hyperparameter optimization method won’t work very well for larger learning rate. Does that agree with your experiment results, and any ideas on how to solve this issue?
>
> Thank you for your valuable question. For the case of large learning rate, it is better to use discrete modeling as shown in (https://arxiv.org/abs/2012.03636). For intermediate learning rates, there are continuous and discrete transitions, so the proposed model can predict up to a certain extent. Also, as you said, the proposed hyperparameter search may not work well when the learning rate is large. However, in our experiments, we have limited the search range of hyperparameter to the effective range based on figure 1. Therefore, we believe that the answer to the former question is consistent with the experimental results. As for the answer to the latter question, we can solve this problem by limiting the effective hyperparameter range by performing Learning Rate Range Test (LRRT) or other tests in advance.
>
> > The model treats terms that contain t as constants (e.g. a0, a1, …, a4). This seems to assume the total number of training epochs to be fixed. However, in practice, early stopping is often needed for better generalization. How would the proposed method deal with potential early stopping?
>
> Thank you for your valuable question. As you said, we consider the number of epochs as a constant in this study, and if you look at Appendix B.3 and B.4, you will see that the parameters of the proposed model (e.g., a0, a1, ..., a4) converge exponentially to zero or some constant for t. Therefore, it is expected that the model will be more flexible and can be used for different epochs by modeling them closely. In addition, the above modeling will enable the proposed model to handle ealy stopping as well, and we will be able to investigate how ealy stopping is effective for better generalization. Although this research has not been able to investigate to that extent, I think it is a good direction for future work.

---

> > ### Comment · Reviewer_sp7W · 2021-11-22
> > **Thank you for your response**
> >
> > I would like to thank the authors for their response. However, I am leaving my score unchanged at weak reject.
> >
> > The main issues with this work are the theoretical soundness and the disconnection between the theory and practice. Details like the failure to account for time-varying A and the discretization with large learning rates make me question the overall theoretical soundness. On the other hand, the heuristic functional form used in the experiments isn't much connected to the theory (the authors haven't properly addressed questions about this from reviewers GHoM and 8e8R).
> >
> > Overall, this is an interesting piece of work aimed at an ambitious problem, but it needs improvements before publication.

---

### Decision · Program_Chairs · 2022-01-20

**Decision:**

Reject

**Comment:**

This paper derives a PAC-Bayes generalization bound for SGD and uses the results to postulate a functional form for the generalization error as a function of the ratio of the learning rate to the batch size. This functional form is then leveraged to develop a kernel function GP hyperparameter optimization.

The reviewers favorably viewed the novel PAC-Bayes bound, but were not convinced by the subsequent analysis. In particular, the reviewers expressed some skepticism about the soundness and generality of the proposed functional form, and were unconvinced that the method would be useful in practice. As such, I cannot recommend the paper for acceptance.